# Trilobite compound eyes with crystalline cones and rhabdoms show mandibulate affinities

Gerhard Scholtz[1], Andreas Staude[2,4] & Jason A. Dunlop[3]

Most knowledge about the structure, function, and evolution of early compound eyes is based on investigations in trilobites. However, these studies dealt mainly with the cuticular lenses and little was known about internal anatomy. Only recently some data on crystalline cones and retinula cells were reported for a Cambrian trilobite species. Here, we describe internal eye structures of two other trilobite genera. The Ordovician *Asaphus* sp. reveals preserved crystalline cones situated underneath the cuticular lenses. The same is true for the Devonian species *Archegonus* (*Waribole*) *warsteinensis*, which in addition shows the fine structure of the rhabdom in the retinula cells. These results suggest that an apposition eye with a crystalline cone is ancestral for Trilobita. The overall similarity of trilobite eyes to those of myriapods, crustaceans, and hexapods corroborates views of a phylogenetic position of trilobites in the stem lineage of Mandibulata.

[1] Humboldt-Universität zu Berlin, Institut für Biologie/Vergleichende Zoologie, Philippstr. 13, 10115 Berlin, Germany. [2] Fachbereich 8.5 "Mikro-ZfP", BAM Bundesanstalt für Materialforschung und –prüfung, Unter den Eichen 87, 12205 Berlin, Germany. [3] Museum für Naturkunde, Leibniz Institute for Evolution and Biodiversity Science, Invalidenstr. 43, 10115 Berlin, Germany. [4] Present address: Thermo Fisher Scientific, c/o Zuse Institut Berlin (ZIB), Takustr. 7, 14195, Berlin, Germany. Correspondence and requests for materials should be addressed to G.S. (email: gerhard.scholtz@rz.hu-berlin.de)

The characteristic facetted compound eyes of euarthropods that are composed of a number of visual units called ommatidia is certainly one of the key characters for the extreme diversification of this animal group[1–3]. Hence, there is an enormous body of literature dealing with developmental, morphological, physiological, and evolutionary aspects of this important sense organ[1–3]. Among Recent euarthropods two major types of compound eyes occur: (1) those that possess ommatidia with a dioptric apparatus comprising a cuticular lens and a cellular crystalline cone[4,5] and (2) those that have a cuticular lens with a cone-like extension, fulfilling a similar purpose of collecting light and guiding it to the retinula cells[6,7] (Fig. 1). The former type occurs in mandibulates, namely some myriapods, crustaceans, and hexapods, whereas the latter type is characteristic for chelicerate horseshoe crabs. Accordingly, it has been frequently suggested that a crystalline cone made up of four cone cells is an apomorphic character for Mandibulata or at least for Tetraconata (crustaceans and hexapods)[2,5,8–12].

The origin of compound eyes dates back at least to the Lower Cambrian and there are a number of fossils from the early euarthropod stem lineage for which the existence of compound eyes has been documented[3,13–15]. Nevertheless, most details about fossil compound eye structures stem from investigations on trilobites[16–18]. However, these reports relate mainly to the cuticular parts of the eyes, i.e., the lenses, whereas soft parts have only rarely been conserved[19]. This situation hampered the classification of trilobite eyes with respect to the modern euarthropod eye types.

Recently, the occurrence of crystalline cones has been suggested for the early Cambrian trilobite *Schmidtiellus reetae*[20]. Yet, the eye of this olenelloid specimen shows a somewhat unusual pattern when compared with the compound eyes of other trilobites and those of modern mandibulates. No other known trilobite species shows such extended and flat cuticular 'lenses'. Likewise, the combination of these 'lenses', the steeply pointed

triangular shape of the putative crystalline cones, the great distance between the ommatidia, and the basket-like structure that encloses each ommatidium[20] finds no correspondence among any mandibulate group. Hence, some doubts remain and in order to come to firmer conclusions about trilobite eyes and their relation to the eye types of other euarthropods further data on internal structures of trilobite ommatidia are required.

Here, we report findings about the internal structures of trilobite eyes using techniques such as Synchroton X-rays, μ-CT, and SEM. We reinvestigate preparations of an Ordovician asaphid trilobite made by one of the pioneers of trilobite eye research, Gustaf Lindström, more than 100 years ago[21]. In addition, we study a newly collected Devonian proetid trilobite from the Eifel in Germany. We provide direct evidence for trilobite eyes being of the mandibulate type, possessing a crystalline cone in addition to a cuticular lens. Furthermore, we describe fossil preservation of a longitudinal section through a rhabdom indicating that early euarthropod compound eyes possessed the same type of light receptors as those of some modern euarthropods. We suggest that a crystalline cone is ancestral within trilobites. Depending on further data on the eye type of stem lineage euarthropods, our results may corroborate the proposed close phylogenetic relationship of trilobites to Mandibulata.

## Results

**An asaphid eye with crystalline cones.** The surface of the eye of an asaphid (*Asaphus* sp.) from the Ordovician of Sweden in the Lindström collection[21] shows the characteristic convex, hexagonal facets of trilobite holochroal eyes[17,18] (Fig. 2a, b). Underneath each facet is an area with a round cross-section, resembling cross-sections through crystalline cones of Recent mandibulates (Fig. 2c–e). In some cases these round structures are filled with matrix, but in other regions they are occupied by a translucent material (Fig. 2c, e). Transverse sections through the eye of *Asaphus* sp. show that the translucent material is cylindrical to cone-shaped with a rounded internal end (Fig. 2g). The spaces between the cones in the transition area to the hexagonal facets are also filled with calcite, forming rings surrounding either fossilized cones or matrix in cases where the cones were not preserved (Fig. 2c–g). Based on their irregular crystalline structure when compared with the layer of the lenses, they are most likely of diagenetic origin (Fig. 2g). Perhaps these ring projections replaced the pigment cells that optically isolate the cones, as is known from modern euarthropods (Fig. 2d–h)[4,5,7].

To test whether the superficial lens-like structures are original or of diagenetic origin, we compared the surface of the eyes with other cuticular structures on the body of *Asaphus* sp. using SEM. The surface of the head shield is smooth and characterized by small pores indicating setae (Fig. 2i). This shows that it represents fossilized cuticle rather than a diagenetic layer. The transition between the cuticle of the eye and the body (Fig. 2j) and the fact that in some cases the putative crystalline cones are displaced or missing (Fig. 2c–g) suggest that these cones are not part of the cuticle as is the case for the cones in xiphosurans. A comparison with a fracture through the exuvia of a horseshoe crab reveals these differences (Fig. 2k). Here the cuticular lens cones are part of the endocuticle, which is smooth outside the eye region. The exocuticle of the hexagonal lenses and the body form a layer of similar thickness lying on top of the endocuticle. In conclusion, the combination of a hexagonal outer cuticular lens with a separate cone-shaped inner structure strongly suggests that *Asaphus.* sp. possessed a cuticular lens in combination with a crystalline cone, as is found in Recent mandibulates (Figs. 1, 2d, h).

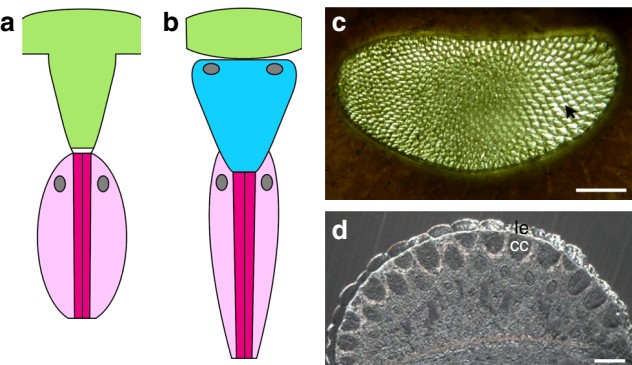

**Fig. 1** The two major compound eye types of Recent euarthropods. **a**, **b** Simplified schemes of the dioptric apparatus of a single unit (ommatidium) of a Recent xiphosuran eye (**a**) and a Recent mandibulate eye (**b**). Pigment cells are omitted. Light green: cuticular lens, turquoise: crystalline cone, white: vitreous cells, dark magenta: rhabdom, light magenta: retinula cell bodies gray: cell nuclei. **a** The cuticular lens forms a cone-like extension. The light is guided via a small transparent vitreous region of about a hundred cells to the rhabdom, i.e. the light-perceiving microvilli of the circularly arranged retinula cells. **b** A relatively flat cuticular lens is combined with a cellular transparent crystalline cone. The position of the nuclei of the crystalline cones and the retinula cells differ among the mandibulate taxa. **c** Micrograph of the internal view of a compound eye of the horseshoe crab *Limulus polyphemus* with numerous cuticular cone-like projections (arrow). **d** cross-section through the eye of the centipede *Scutigera coleoptrata* showing biconvex cuticular lenses (le) and the cellular crystalline cones (cc). Scale bars: 200 μm (**c**), 50 μm (**d**)

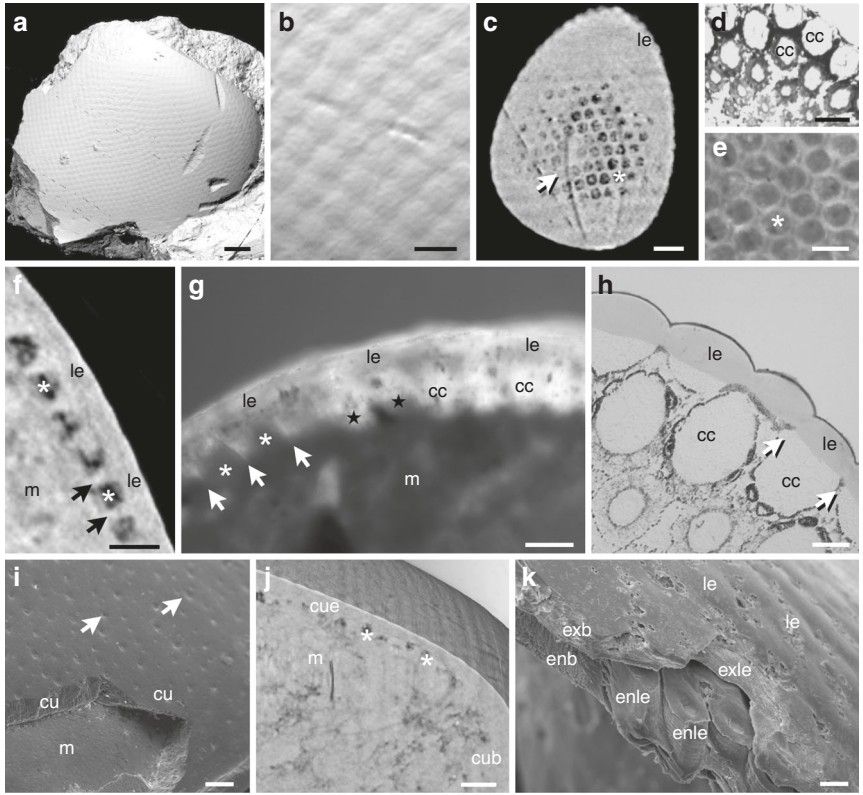

**Fig. 2** The compound eye of *Asaphus* sp. compared with those of a Recent centipede and xiphosuran. Trilobite images show specimens of the Lindström collection (see Pl. 1, Figs. 28, 30 of Lindström[21]). **a, b, c, f, i** Synchroton scans of Ar0019635. **e, g** Semi-thin sections of Ar0059402, **j** SEM of Ar0019635. **a** Surface rendering of the *Asaphus* eye. **b** Close up of **a** showing the convex hexagonal facets. **c** Tangential section of **a** underneath lenses with putative crystalline cones, indicated by dense filling (arrow), missing cones indicated by dark areas (asterisk). **d** Cross-section through round crystalline cones surrounded by dark pigment cells of the centipede *Scutigera coleoptrata*. As in *Asaphus* these lie underneath hexagonal lenses. **e** *Asaphus* tangential section. The white rings correspond to the pointed projections in **g**. **f** Transverse section of **a**. Asterisks mark absent cones, arrows mark ring-like processes. **g** Cross-section through the eye of *Asaphus* showing lenses, crystalline cones, displaced crystalline cones (stars), and missing crystalline cones (asterisks). Pointed projections (arrows) form rings surrounding the empty spaces (see **e**, **f**). **h** Transverse section of a *S. coleoptrata* eye with cuticular lenses and cellular crystalline cones. Arrows point to areas that correspond to the ring-like projections shown in **d–g**. **i** The fossilized cuticle of *Asaphus* with pores (arrows). **j** Section mode (cross-section) of **a** showing the transition between the thin cuticle of the eye and the thick cuticle of the body. Asterisks mark absent crystalline cones. **k** Fracture of the exuvia of the xiphosuran *Limulus polyphemus*. The cone-like projections of the round to hexagonal eye-lenses are part of the endocuticle. The endocuticle of the body is smooth. The exocuticle of the lenses and the body region lie on top. Scale bars 200 μm (**a**, **i**), 50 μm (**b**), 80 μm (**c**), 30 μm (**d**), 40 μm (**e–g**), 20 μm (**h**), 100 μm (**j**), and 60 μm (**k**). cc, crystalline cones; cu, cuticle; cub, cuticle of body; cue, cuticle of eyes; enb, endocuticle of body; enle, endocuticle of lenses; exb, exocuticle of body; exle, exocuticle of lenses; le, lenses; m, matrix

**A proetid eye with crystalline cones and rhabdoms**. The exceptionally preserved specimen of *Archegonus warsteinensis* from the Upper Devonian of Germany stems from a limestone bed containing many disarticulated body parts of the same species (Fig. 3). The eye is broken along its short axis and reveals an almost perfect transverse section (Fig. 3a–c). The cuticle of the eye region exhibits hexagonal biconvex lenses (Fig. 3b). Underneath the lenses, layered calcite forms elongate structures tapering towards the inner region with a fan like arrangement (Fig. 3a, c). A μ-CT scan reveals that these are three dimensionally arranged independent units and not just a superficial result of the fracture of the rock (Fig. 3b inset). Thus, this pattern is strongly reminiscent of the ommatidial organization of Recent compound eyes. A closer view reveals that this similarity concerns even more intriguing details. Adjacent to the lenses there are cone-shape structures in the same position as the crystalline cones of fossil and Recent crustaceans (Fig. 3c–f)[4,22]. As in *Asaphus* sp. some optical units lack the cone-like structure underneath the cuticular lenses (Fig. 3f). This shows that lenses and cones are separate dioptric elements and it is further evidence for the mandibulate nature of trilobite eyes. Underneath each of these putative crystalline cones run long and narrow layered bands with a central axis, which strongly resemble the light-perceiving rhabdom of modern compound eye retinula cells (Fig. 3e, g). Not only the arrangement and the shape, but also the size classes agree between these fossil structures and corresponding modern compound eye components. For instance, the putative rhabdom of the trilobite eye has a diameter of 15 μm (Fig. 3e, g), which exactly matches that of the *Meganyctiphanes norvegica* crustacean rhabdom.

Furthermore, the fact that the putative trilobite rhabdom begins close to the crystalline cones suggests that the eye of *Archegonus warsteinensis* is of the apposition mandibulate type (Fig. 3e). Superposition eyes are characterized by a certain distance between the crystalline cones and the rhabdom, the so-called clear zone[4] (Fig. 3d). At first sight the preservation of delicate structures such as microvilli of a rhabdom seems very unlikely. However, there are cases of unexpected fossil details[23]. In particular, the preservation of the internal eye anatomy of the Jurassic thylacocephalan crustacean *Dollocaris ingens* reveals many details including crystalline cones that resemble the pattern found in the trilobites studied here[22]. Moreover, the complex

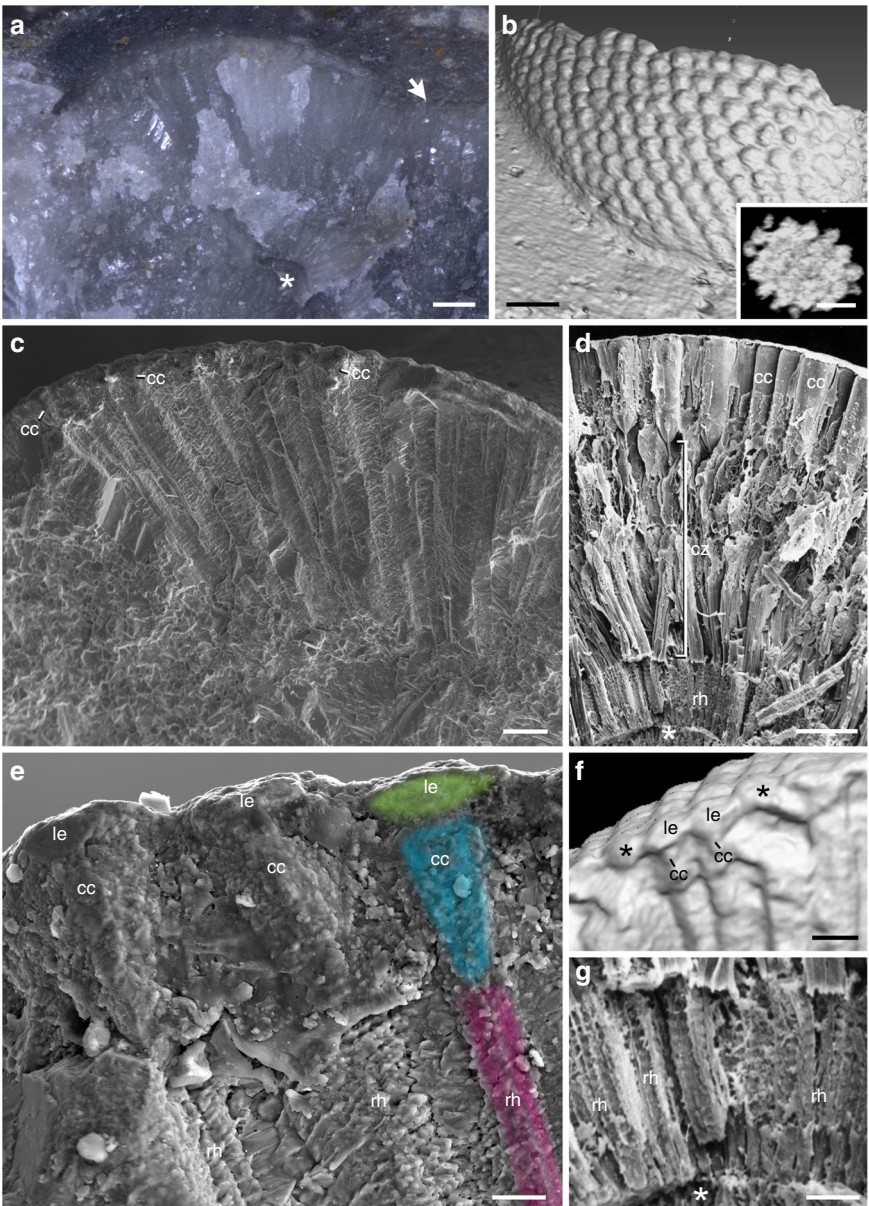

**Fig. 3** The compound eye of *Archegonus warsteinensis* compared with a Recent crustacean eye. Microphotograph (**a**), μ-CT scan (**b, f**), SEM (**c–e, g**). **a** Inner perspective of the eye of *A. warsteinensis* with the regular arrangement of calcite underneath the cuticular facets. Arrow marks the transition between the eye and body, the cuticle thickens and the calcite shows no regular pattern. Asterisk marks conserved part of the compound eye's basal membrane. **b** Surface rendering showing the hexagonal facets. Inset: Section tangential to the eye's surface revealing the separate putative optical units. **c** Similar perspective as in (**a**). The regular pattern of the internal eye parts is visible. Each elongate structure underneath a facet is subdivided into distinct elements, which putatively represent ommatidia. **d** For comparison, the fracture of an eye of the crustacean *Meganyctiphanes norvegica* displaying the ommatidial structure: crystalline cones and retinula cells with layered rhabdoms (lenses removed) (after ref. [4], with permission of the author). Between the crystalline cones and the rhabdom there is a clear zone characteristic of refractive superposition eyes[4]. Asterisk marks the compound eye's basal membrane (compare with **a**). **e** Enlarged part of (**c**) depicting the details of three putative ommatidia of the fossil trilobite eye. The biconvex lenses (light green) are combined with cone-shaped crystalline cones (turquoise) and adjacent elongate rhabdoms (magenta) resembling those of modern euarthropods (compare with **f, g**, and Fig. 1b). **f** Surface rendering of the lens - crystalline cone complex of another part of the eye of *A. warsteinensis*. The asterisks mark two lenses lacking crystalline cones showing the biconvex form of the lenses and indicating that cones and lenses are separate structures. **g** The rhabdom of *M. norvegica* (enlarged from **d**) revealing a corresponding structure to that of *A. warsteinensis* (after ref. [4], with permission of the author). Asterisk marks the basal membrane of the eye. Scale bars 200 μm (**a**), 150 μm (**b**), 100 μm (**b** inset, **c**), 50 μm (**d, f**), 20 μm (**e**), and 10 μm (**g**). cc, crystalline cones; cz, clear zone; le, lens, rh, rhabdom

pattern found in the eye of *Archegonus warsteinensis* renders it unlikely that the similarity between the fossilized structures and Recent compound eye elements is based on diagenetic processes accidentally mimicking organismal structures. Instead, the growing calcite crystals are presumed to replace the soft internal eye parts and assumed their shape.

## Discussion

Some authors previously suggested that trilobites possessed crystalline cones, albeit without direct evidence[3,24], while others analyzed the function of the lenses without taking other putative optic elements into account[16]. Only recently was evidence reported for crystalline cones in the Cambrian olenelloid trilobite

*Schmidtiellus reetae*[20]. This specimen is phosphatized and the surface of one eye is broken revealing internal elements such as a cone-like structure and a rosette-like arrangement of small round structures. These were interpreted as putative crystalline cones and retinula cells respectively. Moreover, it was suggested that the eye was of the apposition type[20]. However, in fossils it is difficult to discriminate between a cellular crystalline cone and a cone-like-extension of the cuticle. Hence, we used the approach of the process of elimination to show that in trilobite eyes we actually deal with crystalline cones. A cuticular cone as is found in xiphosurans is confluent with the outer part of the cuticle, whereas the crystalline cone of mandibulates is an independent morphological unit. The absence of cones in some of the ommatidia of the two trilobite species studied by us in combination with an unaffected inner lens surface strongly suggests a morphologically independent cone structure, and thus the existence of proper crystalline cones. Likewise, the existence of rhabdoms in *Schmidtiellus reetae* and other trilobites[19,20] was so far only indirectly inferred based on cell-like structures that resemble the characteristic circular arrangement of retinula cells. Our study presents the first direct evidence for the microvilli structure of a rhabdom in a trilobite apposition eye and thus provides additional and clear evidence confirming the existence of a mandibulate eye type in trilobites.

The internal phylogeny of trilobites is far from being settled[25,26]. Nevertheless, most hypotheses of trilobite phylogeny place Olenellina or parts thereof as sister group to the remaining trilobites[25,26]. Furthermore, Asaphida and Proetida are resolved as being nested within the trilobite tree[25,26]. Hence, the occurrence of crystalline cones in representatives of these three groups allows the tentative conclusion that the ancestral trilobite eye was equipped with crystalline cones like the eyes of modern day mandibulates. In addition, it is likely that this ancestral eye was of the apposition type. However, the eye of *Schmidtiellus reetae* differs in several aspects from those of *Asaphus* sp., *A. warsteinensis*, and modern mandibulates. This relates to the few and distantly arranged ommatidia, the relatively large but flat cuticular 'lenses', the small, steeply pointed crystalline cones with straight margins, and the internal structures of the eye being enclosed by a basket-like structure of unknown histology[20]. Due to the great geological age of *Schmidtiellus reetae*, these specific characters have been suggested as primitive for compound eyes[20].

However, several lines of evidence cast some doubt about this conclusion. The genus *Schmidtiellus* is deeply nested within the Ollenelloidea[27] and other olenellid species show proper biconvex lenses that are densely packed[28,29]. Eyes with numerous hexagonal lenses are also found among xandarellids, close relatives of Trilobita[30]. Moreover, the compound eyes of radiodontans, one of the early branches of the euarthropod stem lineage, possessed eyes with hexagonal lenses in a dense arrangement[13]. Finally, a stratigraphic earlier age does not necessarily mean that characters show an ancestral state[31]. Thus, given the correspondence between the overall eye morphology of *Archegonus warsteinensis* to fossil and modern mandibulates, the peculiarities of the eyes of *Schmidtiellus reetae* might in fact be evolutionarily derived, despite its great age. This suggestion has precedent among Recent mandibulates. For example, several crustacean taxa such as branchiopods, leptostracans, and amphipods possess crystalline cones but lack cuticular lenses[2,32,33]. At least for Amphipoda this is clearly apomorphic, since they are deeply nested within malacostracans, which have cuticular lenses of different shapes[2,4,6,34,35]. Similarly, apomorphically reduced and strangely formed crystalline cones can be found, for example, among isopods, brachyuran crabs, penicillate myriapods, and wingless hexapods[11,35–37]. However, none of these species show the great

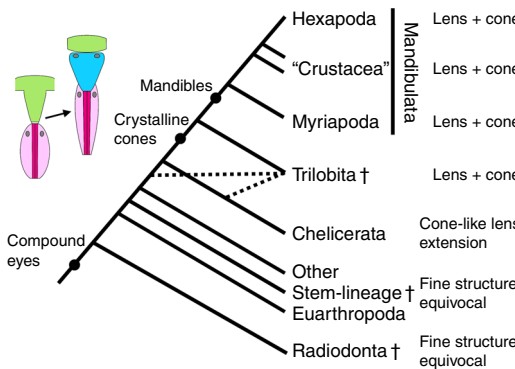

**Fig. 4** Simplified cladogram of stem-lineage and crown-group of Euarthropoda (Mandibulata + Chelicerata). The evolutionary key characters of eye evolution are shown (combined from several articles[38–41]). The 'other stem-lineage Euarthropoda' largely comprise the taxa that are sometimes considered as 'upper-stem Euarthropoda'[46]. Compound eyes with crystalline cones likely evolved in the lineage leading to Trilobita and Mandibulata comprising Myriapoda, Hexapoda, and paraphyletic crustaceans, rendering trilobites part of the mandibulate stem-lineage. Alternative hypotheses about the phylogenetic position of trilobites (stem lineage of Chelicerata[39] or stem-lineage of Euarthropoda[38]) are indicated by dotted lines. If crystalline cones were already present in the compound eyes of Radiodonta[3], then this character would not be informative with respect to the phylogenetic position of Trilobita. However, this changes to the opposite case if some of the other stem-lineage euarthropods that are more closely related to crown-group euarthropods possessed ommatidia with cuticular cones, as is likely the case for megacheirans[3]

interommatidial distance and the basket-like structure of the compound eyes of *Schmidtiellus reetae*.

The phylogenetic position of trilobites within the euarthropods is still elusive. There are indications for a position in the euarthropod stem lineage[38], a close relationship to chelicerates[39] and likewise for a mandibulate affinity[40,41]. The general view on euarthropod eye evolution is that the differentiation of a crystalline cone made up of four cone cells is an evolutionary novelty of some myriapods and Tetraconata or even an apomorphy of Mandibulata[10–12]. Hence, the detailed correspondence between the patterns of trilobite and mandibulate compound eyes strongly suggests homology and may corroborate a close relationship of trilobites to myriapods, crustaceans, and hexapods (Fig. 4). Yet, in a recent publication the view of crystalline cones being a mandibulate apomorphy has been challenged. Based on investigations on compound eyes of a number of putative Cambrian stem lineage euarthropods, it was suggested that crystalline cones already occurred in the earliest known facetted eyes of the Radiodonta. Accordingly, the absence of crystalline cones as in Xiphosura was interpreted here as the result of an evolutionary reduction or loss[3].

However, this study leaves some questions open. As mentioned above, in fossils the discrimination between cuticular cones and cellular crystalline cones is not straightforward and requires a more scrutinized approach than that executed by the authors of this study[3]. Furthermore, any evolutionary scenario of euarthropod compound-eye evolution depends on the phylogenetic relationships of the various taxa in the euarthropod stem lineage. Yet, these relationships are still very controversial. In particular, this is true for the content and phylogenetic position of artiopods, megacheirans, and fuxianhuiids. These groups have been alternatively resolved as stem-lineage euarthropods or within crown-group Euarthropoda—e.g., megacheirans as ancestral chelicerates, artiopods with or without chelicerates, and fuxianhuiids as mandibulates[3,38,39]. Thus, any reconstruction of the evolution of

the different compound eye patterns is necessarily biased by this situation. For instance, if the megacheiran *Leanchoilia illecebrosa* possesses a chelicerate eye-type, as has been suggested[3] and if it is a stem lineage euarthropod[38,39], then this renders the structure of the ommatidia of the xiphosuran eye plesiomorphic for crown-group euarthropods.

Hence, the well-founded structural correspondences between the eyes of trilobites and mandibulates may indeed presently favor a close relationship between these groups and Trilobita might be part of the mandibulate stem-lineage, as previously suggested[40–42]. Moreover, since data on early crystalline cones in early Cambrian euarthropods remain somewhat equivocal and open to alternative interpretations, it would be interesting to confirm whether the compound eyes of Radiodonta and other Cambrian stem-lineage euarthropods show the xiphosuran or mandibulate eye type. What is needed is a systematic screening and detailed investigation of the preserved compound eyes of these groups. Together with a better resolution of the phylogeny of euarthropod stem lineage taxa, this would clarify the direction of euarthropod eye evolution and contribute towards the understanding of evolutionary pathways in these crucially important organ systems.

## Methods

**Fossil material**. We used the original trilobite material from Gustaf Lindström housed in the Naturhistoriska Riksmuseet, Sektionen för Paleozoologi, Stockholm (Sweden). The material comprises a number of microscopic preparations of the eyes of various trilobite species among them two of an undetermined asaphid specimen, *Asaphus* sp. from the Ordovician (Gotska sandön, Gotland, Sweden) (Ar0059402). From this specimen the anterior part exists from which Lindström had cut off parts for microscopic preparations (Ar0019635). According to the labels of the Riksmuseet this is a Silurian species. Yet, the structures of the anterior head clearly identify this specimen as an asaphid, a group that became extinct by the end of the Ordovician[43]. In addition, specimens of *Archegonus* (*Waribole*) *warsteinensis* from the upper Devonian (Fammenian) of Germany (Kalvarienberg/Kallenhardt) were collected by Dieter Korn. This material has been deposited in the collections of the Museum für Naturkunde (MB.T 7303), Berlin, Germany.

**Recent animals and histology**. Fixed specimens of *Scutigera coleoptrata* from the Zoological Teaching Collection of the Humboldt-Universität zu Berlin were used for eye histology. Recent animals were embedded into methacrylate (Kulzers Technovit®) and semi-thin sectioned (2–3 µm) (Microtome Zeiss HM 2165). The sections were transferred onto microscopic slides and stained with a mixture of methylene blue and Azur-II (1% methylene blue in 1% aqueous borax solution, 1% Azur-II in aqua dest., 1:1) for about 5 min.

**Imaging**. High-resolution X-ray computed tomography measurements were performed at the synchrotron-beamline BAMline (BESSY II, Helmholtz-Zentrum Berlin für Materialien und Energien[44]) and at the BAM 225kV-µCT device. For synchrotron measurements, a monochromatic energy of 25 keV was used. Using the Princeton Instruments camera (VersArray: 2048B), the voxel-size of the datasets is 3.7 µm. 1200 projection images were taken for a 180 degrees rotation (which gives complete information in parallel-beam geometry). The BAM 225 kV-µCT device featured a microfocus X-ray tube (manufacturer: Feinfocus) with a maximum acceleration voltage of 225 kV, and a PerkinElmer flatpanel detector with 2048 × 2048 pixels. For the measurements, an acceleration voltage of 80 kV was used. A pre-filter of 0.25 mm copper was applied. 1500 projections were taken for a full rotation. The voxel-size was 4.8 µm. Both, the synchrotron and the µCT data, were reconstructed using a filtered-backprojection algorithm[45]. For visualisation and analysis, we used VGStudio Max 2.0 and Amira 5.4.3 software.

**Scanning electron microscopy**. Exuviae of *Limulus polyphemus* collected from beaches in Cold Spring Harbor (USA) and the fossils were mounted on a stub and sputter coated with gold (Balzers Union). Observation and micrographs were done with a LEO 1450 VP scanning electron microscope. The sections and the Lindström preparations were analyzed with a light microscope (Zeiss Axioskop 2 plus) using mostly differential-interference-contrast and photographed with an Axio-Cam HRc equipped with the software AxioVision 4.8. Microphotographs were done with stereomicroscopes equipped with digital cameras (Keyence VHX-1000 and Leica MZ16 with Sony PMW-10MD). Figures were compiled and global contrast and brightness values of some of the images were adjusted using the Ulead Photoimpact 12 software.

**Reporting summary**. Further information on research design is available in the Nature Research Reporting Summary linked to this article.

## Data availability
The datasets generated during and/or analyzed during the current study are available in the Humboldt-Universität zu Berlin: edoc-server repository, https://doi.org/10.18452/20002.

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

## Acknowledgements

We thank Renate Mbacke, Kristin Jütz (histology), Peer Martin, Thomas Stach, Juliane Vehof (SEM), and Kristin Mahlow (μ-CT) for technical support. The help of Dieter Korn, who collected and donated the specimens of *Archegonus warsteinensis* and of Stefan Richter, who prepared and SEM-photographed the Recent crustacean eyes (Fig. 3d,g), is gratefully acknowledged. Thank is due to the late Jan Bergström, for the loan of the Lindström collection (Stockholm) and to Dan Nilsson, Dieter Korn, Michael Steiner, and Stefan Richter for discussions. This research was funded by the Deutsche Forschungsgemeinschaft, project: 'Structural Flexibility in the Optical Design of the Arthropod Cornea' (Scho442/15–1).

## Author contributions

G.S. designed the study, conducted most experiments, interpreted the data, and wrote the first draft of the manuscript. A.S. performed the μCT and synchroton CT measurements. J.D. interpreted and discussed the results. All authors contributed in writing the manuscript and commented on the manuscript at all stages.
