## [Peer Review File · Nature Communications]

Reviewers' Comments:

Reviewer #1:

Remarks to the Author:

This is a valuable contribution to anatomical knowledge of one of the "poster children" of palaeontology, trilobites. The sublensar anatomy of trilobite eyes has been exceedingly elusive, so collecting data on the crystalline cone and rhabdom allows trilobites to be incorporated into arthropod phylogeny based on eye characters. The paper requires little revision to be published.

I would like to see the images in Fig. 1j and 2f without the limits of one of the dioptric apparatuses being covered by suggestive outlines. In Fig. 1j, the precise one-to-one correspondence of lenses and crystalline cones shown in the outline is less clear in all of the other ommatidia. Perhaps the authors covered up the most compelling example. Likewise in Fig. 2f, only three ommatidia are shown and one of them is covered by the outline so we lose the ability to see if the lens really is biconvex and really is separated from the crystalline cone. I think the authors are probably right in both cases but wish they hadn't reduced the amount of evidence we have to evaluate the argument.

Figure 2 could use a bit more labelling. Image d should label the rhabdom (rh) for more direct comparison with Fig. 2e. Image e should have the clear zone labelled.

Line 101 would be an appropriate place to cite a rather surprising omission. Vannier et al (2016, Nature Communications) documented exquisite preservation of the cellular-scale anatomy of eyes of Jurassic thylacocephalan crustaceans, including the crystalline cone and details of reticular cells. One would think this should be noted.

Some trivial editorial corrections follow:

- Line 19: "some myriapods" would be an appropriate wording, as the data come only from Scutigleromorpha and Penicillata;
- Line 50: Fig. 1j should be lower case;
- Line 57: "semi-thin" rather than "semi-thins";
- Line 78: Fig. 1k rather than Fig. 2;
- Lines 80 and 269: Upper Devonian should be upper case, as a formal chronostratigraphic unit;
- Line 81: it is a limestone bed rather than a calcite bed;
- Line 83: Fig. 2b rather than Fig. 4b.
- Figure 1i is lettered in upper case;
- Line 151: "from" rather than "form";
- Line 153: although in this simplified tree Trilobita "is" sister-group of Mandibulata, it might be better to say that trilobites are stem-group Mandibulata, to allow for the fact that other stem-group mandibulates (such as Phosphatocopina) are probably more closely related to the mandiblate crown group;
- Line 189: Fig. 2e, g, h rather than Fig. 4 ;
- Line 265: Use the formal name for the "the zoological museum" in Stockholm, e.g., Naturhistoriska Riksmuseet;
- Line 271: Which collection is "the zoological collection"?
-
-

Reviewer #2:

Remarks to the Author:

This account claims to be the first describing the lens and crystalline cone, and indeed rhabdomeres, of a trilobite eye. The evidence is based on extremely poor material that is fancifully interpreted so as to shoe-horn the data as support for an apposition eye conveniently matching that of a malacostracan crustacean. The authors use their interpretation as a vehicle to criticize several recent papers on Lower Cambrian eyes that likewise claim apposition eyes as the ancestral type. It is unfortunate the authors specifically criticize material, on which those papers are based, when their own material is so very ambiguous.

The paper is sloppily written, possibly in haste. illustrations are of low quality. Certain panels in the figures are not described in the text and the fourth figure is entirely absent.

The reasoning and logic used to support their data and deride existing publications is often muddled and without factual basis.

What follows is a line-by-line review of the text.

Lines 10-11. "This is in particular true for the eyes of fossils." But the authors then later negate their own opinion by rejecting the interpretation of the radiodontan eye type.

Line 16. "(1h,j)". The standard procedure is to number consecutively starting from the lowest.

Line 21. ".. "the evidence is equivocal...." not at all: the cited authors present quite respectable evidence, superior to the ambiguous images present in the present manuscript.

Lines 22, 23. "Moreover, the described structures might be in fact cuticular processes as in horseshoe crabs." Not sure what "in fact" is supposed to infer. In any event, this comment is distinctly odd coming from an author who has suggested trilobites are mandibulates.

Line 36. Fig. 1 is a mélange of images, not all from trilobites. To what are the authors referring?

Line 49 "...as the cones in xiphosurans." Should be written: "as are the cones in xiphosurans."

Lines 49-50. ".....a histological section through the exuvia of a horseshoe crab..." This is not a section but a fracture, and the preservation is so poor that the external features of the lenses appear to have collapsed. There is no reference to Fig. 1k, which so long as it's not mentioned remains irrelevant to the description.

Lines 76-77. "To test whether the superficial lens-like structures are original or of diagenetic origin, we investigated other cuticular structures of the body of the fossil using SEM." The reasoning is flawed. Just because one area of cuticle surface appears to have a regular structure does not mean that other areas have been spared diagenetic alteration. Likely diagenetic artefacts are nicely indicated at and beneath the surface of the so-called "lenses" in figure 2f.

Line 78. Fig. 2 is a medley of images, none which show head shield cuticle.

Line 101. "However, there are cases of unexpected fossil details¹³." What relevance has this paper - which described issues pertaining to soft tissue preservation - to do with the present manuscript?

Lines 104, 105. "Rather the growing calcite crystals replaced the soft internal eye parts and assumed their shape." This is an assertion. But as far as one can see there is no experimental evidence to support it.

Lines 123-124. "Taking all this together, we suggest that there is good evidence that these structures indicate the occurrence of the mandibulate eye type in Trilobita." There is no compelling evidence in this manuscript to support this claim. Authors do themselves no favors by citing other papers on the topic of doing less. Indeed, at least one of the cited papers showed evidence that excludes limulus-type optics in trilobites proposing instead that the ancestral trilobite eye was an apposition type. Have the present authors showed anything more substantive? Note really. Their main piece of evidence, Fig. 2f, is replete with ambiguity; drawing a lens and cone onto wholly asymmetric structures is no substitute for actual evidence and is a strategy that could expose the authors to ridicule. Likewise, Fig. 2d, which shows columns suggesting visual units, fails to provide any evidence of crystalline cones; the labels "cc" indicate wish-fulfillment by whichever author placed the labels. Returning to Fig. 2f, even if the outline I and cc was a true indication of the dioptric apparatus, which it is certainly not, there would have to be an explanation as to why the claimed lenses (labeled I) are so enormously far apart from each other. There is no evidence of lenses that should be partly visible, albeit offset, between the fractured lenses (if that is what they are) such as would typify the apposition hexagonal mosaic. Nor does the fractured eye of *N. integer* help, as the preservation is so poor that would the figure legend not have stated what it is, there would be no way of knowing it belonged to a compound eye.

Lines 125, 126. "Some authors suggested that trilobites possessed crystalline cones without direct evidence." This is true. And judging from the claims made in Fig. 2f one could add the present paper to that list. The Schoeneman et al. 2017 paper on the lower Cambrian *Schmidtiellus reetae* thus remains the most convincing description yet. And yes, the ommas are spaced further apart than in an extant apposition eye, but one must consider the status of *Schmidtiellus* as not simply a Cambrian specimen, but one from about 525 mya, thus coeval with *Radiodonta* and other stem euarthropods.

lines 134-137. "Our study on eye structures in other trilobites such as Asaphidae and Proetidae provides additional evidence for the existence of crystalline cones and thus supports the suggestions of Schoeneman et al." That's not very generous is it? Surely it should read "Our study on eye structures in other trilobites confirms the observations of Schoeneman et al." (although, as remarked already, this would be a generous self-appraisal as Figure 2f hardly confirms anything. And the present study is on one trilobite species, so I'm not sure what evidence is shown here for the "other" trilobites.

Lines 137-140. "In addition, the cones in the asaphid and *Archegonus warsteinensis* are related to proper cuticular lenses and the eyes comprise more than thousand ommatidia. Furthermore, we present the first evidence for the existence of a rhabdom in a trilobite apposition eye complementing previous findings on retinula cells." 1. What does it matter how many ommatidia (actually lenses) were counted; 2) and, no, it is Schoenemann et al. who provide the first evidence.

Lines 142-145. "Moreover, trilobite eyes show a structural diversity that is comparable to that of Recent crustaceans. Like *Schmidtiellus reetae* the eyes of branchiopod, leptostracan, and amphipod crustaceans possess crystalline cones but lack cuticular lenses." Three taxa lacking cuticular lenses do not Crustacea make! As Dan Nilsson once remarked, the *Brachyura* alone have more eye types than found across all other crustaceans. So far, four types of trilobite eyes have been described, only two of which consist closely packed facets suggestive of apposition eyes. And why contrast decapods. isopods, stomatopods? These too are recent crustaceans. This reviewer fails to grasp of this.

line 160-166 and 167-170. "Yet, in two recent papers, it was claimed that the compound eyes of the Cambrian arthropod species *Fuxianhuia protensa* (*Fuxianhuiida*), *Fortiforceps*, and *Lyrapax unguispinus* (*Radiodonta*) possessed crystalline cones^{15,25}. Since radiodontan species in particular

have been considered as stem lineage representative of arthropods 2,23, this would mean that the earliest known faceted eyes were already equipped with the complex dioptric apparatus of modern day mandibulates. For several reasons this is hard to accept:..." It is? Only if the authors offer evidence for an alternative, which they do not. Rather, they take issue with the radiodontan fossil eye, the preservation of which shows a period repeat. This contrasts to the (for want of a more apt descriptor) specious crystalline cone and lens drawn onto what is a highly irregular and asymmetric matrix in their Fig. 2f. Theirs is an example of throwing stones when living in a glasshouse, usually unadvised. While this reviewer has no problem with the authors suggesting a "possible" crystalline cone, what is objectionable is constructing what in Germany is known as a "straw man" to amplify an assertion when it should be a modest suggestion. Taken together, the authors' criticism of Schoeneman et al., (2017) as being less convincing is moot when the present authors', own study appears even less persuasive than it did at first. Nor does the radiodontan description alluded to appear any less persuasive than the images in the present Fig. 2f; indeed, the contrary.

Lines 170-172. "There is no indication that the eyes of *Limulus* underwent a reduction of a crystalline cone since there are no cells corresponding to cone cells." I suggest the authors go deeper into the literature and consult W. H. Fahrenbach (1975). The visual system of the horseshoe crab, *Limulus polyphemus*. *Int. Rev. Cytol.* 41, 285-349, which further suggests that the *limulus* eye type is derived from an ancestral apposition organization (as would the Leacholiid eye).

Lines 177-179. "This falsifies hypotheses that xiphosuran eye morphology is related to a 178 certain degree of reduction of an originally more complex situation." Poor Professor Lindström, falsified after 116 years.

Lines 179-183. "Since the data on early crystalline cones in early Cambrian arthropods are elusive, it would be interesting to test whether the compound eyes of Radiodonta, Fuxianhuiida, and other Cambrian stem-lineage arthropods show the xiphosuran eye type or indeed possess crystalline cones. This would clarify the direction of arthropod eye evolution and contribute to the understanding of the evolutionary pathways of these important organ systems." Elusive data; now that is something new this reveiwer. However, in the event the lines above may have got lost in translation, what the authors are saying, in none too subtle a manner, is that "We don't believe all that stuff claiming apposition eyes in radiodontans et al., and, dear reader, neither should you." I'd hoped this kind of arrogance had gone extinct around the mid-70s.

Reviewer #3:

Remarks to the Author:

In this contribution, Scholtz et al. describe in great detail the fine structure of trilobite eyes, with particular emphasis on the discovery of reasonably convincing crystalline cones, and make comparisons with a recent study suggesting the presence of these features in admittedly less well-preserved material from the lower Cambrian (Schoenemann et al. 2017, PNAS). The new morphological evidence is clear and this study represents a valuable contribution towards a better understanding of the evolution of optic systems in early arthropods. I have a number of suggestions, however, that include providing a more balanced account of the discussion of these findings, particularly with regard to the phylogenetic implications for trilobites, and some formatting suggestions. However, these are for the most part necessary to make the message even clearer, and thus I will be happy to support publication of this paper in Nature Communications after the authors make what accounts to a moderate revision.

General comments:

- There is a general, and rather troublesome, formatting problem in that material is described in the results, but there is no material and methods section that explains anything regarding the provenance of these fossils. For instance, half of the main data in the paper (e.g. Figure 1) is based on an "unidentified asaphid from the Lindstrom collection, Stockholm". However, there is no mention of age, geographic origin, geological context, stratigraphic provenance, or even a museum catalogue number. All of this is essential data that should be clearly provided in the main text. I realize that some of this information is provided as supplementary data for some of the figured specimens, but given the brevity of this manuscript I find no reason to not feature it up front in the main text.

Figures.

The morphological detail preserved in the fossils is quite impressive; however, the use of small panels induces some frustration given that they show interesting details but their size does not allow a clear examination. Please reformat the figures to avoid the use of small inset panels (e.g. Figure 1c, f, g; Figure 2c) and instead display all the available data as clearly as possible.

A second suggestion is the inclusion of a reference diagram that informs the reader about the basic organization and diversity of compound eyes in extant mandibulates and chelicerates, as well as a morphological reconstruction of the eye structure of trilobites (both those reported here, and those of Schoenemann et al. 2017), for clarity. Given that most of the photographs are high magnifications, it is occasionally difficult to identify the structures that are being discussed. A summary and comparative diagram would be extremely helpful to better communicate these points and make the message of the study all the more accessible.

Lines 1 – 31. I get the sense that there should be an abstract here, but this reads more like an introduction to the rest of the paper. I am not aware of Nature Communications have an article format that does not include an abstract, so the authors should include the relevant section here to summarize their findings.

Lines 21-23. Please elaborate on how the preserved eye structures in the material of Schoenemann et al. (2017 PNAS) are unusual compared to recent representatives, otherwise this comment comes across as dismissive and not particularly constructive. I realize something on these thread is mentioned in lines 128 – 134, but those do not make it clear why this organization is incomparable with that of extant forms either.

Line 29. The authors suggest that the crystalline cone is ancestral within trilobites. Given that their material is of either unknown stratigraphic provenance (the uncertain asaphid), or of upper Devonian age (Archegonus), or correspond to demonstrably derived trilobite groups (e.g. asaphida, proetida). Based on this sparse data sampling, it is impossible to claim that this type of organization is unequivocally ancestral for trilobites unless their findings agree with those of Schoenemann et al. (2017 PNAS), which belong to a stratigraphically and phylogenetically trilobite. Then, in line 139, the authors state that they actually do agree with Schoenemann et al. (2017) after all, which makes me wonder why they bring up this study earlier as somehow flawed or incorrect in Lines 21 to 23. I recommend to balance the criticism of Schoenemann in the introduction, which gives the erroneous impression to the reader that the data in that paper is somehow fundamentally flawed or unreliable, and that the authors present their findings in the context of the current understanding of trilobite

vision more clearly.

Line 34. Which are the new trilobite finds? This type of lines make it clear why the materials and methods should be part of the main text.

Line 44. Please include a reference for a study on modern arthropods here.

Line 76. Note that it is entirely possible to have a well-preserved exoskeletal cuticle but diagenetically altered eyes in fossil arthropods (e.g. see *Gogglops ensifer* in Siveter et al. 2017, Geological Magazine). Also, what specimens are we talking about here? Based on the following lines I imaging it is *Archegonus*, but if that is the case, this is conveyed rather awkwardly. Please reorganize this section for clarity.

Lines 80 to 105. This descriptive section further makes a point for including an additional figure with diagrams that guide the reader through the preserved morphology, as it is difficult to distinguish some of the features mentioned in the main text (e.g. clear zone in Figure 2e).

Also, does the CT data provide further evidence of the organization of the lenses in the eyes of *Archegonus*? Figure 2f is a bit too suggestive as there is not a perfectly clear-cut morphological distinction between the lens and the crystalline cone on the exposed surface. Is this could be demonstrated with the CT data obtained it may be possible to segment and isolate these elements to obtain a better view of their morphology.

Line 143. The authors mention that trilobite eyes show a structural diversity comparable to that of recent crustaceans. However, it is not particularly clear that they have described structurally different morphologies based on the unnamed asaphid and *Archegonus*. Is this diversity based only on the length of the crystalline cone? If so, again, a diagrammatic reconstruction would help to better convey these nuances, as they are not evidently clear from the figures alone without a side-by-side comparison.

Line 165. It is true that the phylogenetic position of trilobites is still debated, which is probably due to their proximity to the main dichotomy between chelicerates and mandibulates. Given these considerations, the presence of mandibulate-like ocular structures in trilobites can equally be interpreted as symplesiomorphic relative to crown-group Euarthropoda, in line with recent suggestions that the brain outline of *Fuxianhuia* (Ma et al. 2012, Nature; Ma et al. 2015, Current Biology) and some Cambrian bivalved forms (e.g. *Odaraia alata*, see Ortega-Hernandez 2015, Current Biology), has a mandibulate-like morphology. In this context, it is just as likely that the ocular organization of trilobites, as well as other mandibulate-like features, are actually ancestral rather than synapomorphic. I understand if the authors object to his alternative, but they should at least acknowledge that this scenario is equally supported by the available data, rather than focussing the discussion on their preferred hypothesis.

Line 166. Again, to mention that the "... presentation of the data is not convincing..." is a somewhat

hostile but also superficial criticism. Please explain how it is that this data is unconvincing, rather than just claiming it so. For example, I would agree that the preservation of the eyes in *Fuxianhuia protensa* do not allow to recognize extremely fine details of the internal eye organization, such as the presence of crystalline cones, as demonstrated in Figure 3f of Ma et al. 2012, Nature.

Line 177. Following from the former comment, the authors mention that the eye structure of eurypterids demonstrates that there has been no loss or reduction of the crystalline cone in Xiphosura. However, none of the references cited provide information on the internal organization of the eyes in eurypterids, and as far as I am aware this information is not directly available from fossils, but instead focus on the superficial arrangement of eurypterid eyes and how that informs their ecology. Thus, there is no such demonstration that the eye organization of xiphosurans cannot possibly be secondarily derived. In this context, it is noteworthy that lateral eyes have been gained and lost repeatedly throughout xiphosuran evolution (e.g. see phylogeny in Lamsdell 2016, Paleontology), and thus it is entirely expected that these will represent highly plastic structures. The claim that the eyes of extant xiphosurans reveal the ancestral condition of arthropods is clearly focused on extant diversity alone, and thus probably best discussed with a pinch of salt.

Reviewers' comments:

We thank the reviewers for the careful reading and the valuable comments. Due to their help we feel that our manuscript has been very much improved. We took almost all their comments into account. What follows is a point by point reply to the issues raised by the three reviewers.

Reviewer #1 (Remarks to the Author):

This is a valuable contribution to anatomical knowledge of one of the “poster children” of palaeontology, trilobites. The sublensar anatomy of trilobite eyes has been exceedingly elusive, so collecting data on the crystalline cone and rhabdom allows trilobites to be incorporated into arthropod phylogeny based on eye characters. The paper requires little revision to be published.

I would like to see the images in Fig. 1j and 2f without the limits of one of the dioptric apparatuses being covered by suggestive outlines. In Fig. 1j, the precise one-to-one correspondence of lenses and crystalline cones shown in the outline is less clear in all of the other ommatidia. Perhaps the authors covered up the most compelling example. Likewise in Fig. 2f, only three ommatidia are shown and one of them is covered by the outline so we lose the ability to see if the lens really is biconvex and really is separated from the crystalline cone. I think the authors are probably right in both cases but wish they hadn't reduced the amount of evidence we have to evaluate the argument.

We deleted the outlines in the two figures. The lens structure Fig. 1 (now Fig. 2) becomes clear, when all images of the plate are taken into account. In Fig. 2, (now Fig. 3) we used a transparent colouring instead and we submit the original figure as supplementary material.

Figure 2 could use a bit more labelling. Image d should label the rhabdom (rh) for more direct comparison with Fig. 2e. Image e should have the clear zone labelled.

We added more labels as requested.

Line 101 would be an appropriate place to cite a rather surprising omission. Vannier et al (2016, Nature Communications) documented exquisite preservation of the cellular-scale anatomy of eyes of Jurassic thylacocephalan crustaceans, including the crystalline cone and details of reticular cells. One would think this should be noted.

We agree, the Vanier et al. paper has to be cited here. Now it is included.

Some trivial editorial corrections follow:

- Line 19: “some myriapods” would be an appropriate wording, as the data come only from Scutigromorpha and Penicillata;

done

- Line 50: Fig. 1j should be lower case;

done

- Line 57: "semi-thin" rather than "semi-thins";

corrected

- Line 78: Fig. 1k rather than Fig. 2;

corrected

- Lines 80 and 269: Upper Devonian should be upper case, as a formal chronostratigraphic unit;

corrected

- Line 81: it is a limestone bed rather than a calcite bed;

done

- Line 83: Fig. 2b rather than Fig. 4b.

corrected

- Figure 1i is lettered in upper case;

corrected

- Line 151: "from" rather than "form";

corrected

- Line 153: although in this simplified tree Trilobita "is" sister-group of Mandibulata, it might be better to say that trilobites are stem-group Mandibulata, to allow for the fact that other stem-group mandibulates (such as Phosphatocopina) are probably more closely related to the mandiblate crown group;

done

- Line 189: Fig. 2e, g, h rather than Fig. 4 ;

corrected

- Line 265: Use the formal name for the "the zoological museum" in Stockholm, e.g., Naturhistoriska Riksmuseet;

done

- Line 271: Which collection is "the zoological collection"?

specified-

-

Reviewer #2 (Remarks to the Author):

This account claims to be the first describing the lens and crystalline cone, and indeed rhabdomeres, of a trilobite eye. The evidence is based on extremely poor material that is fancifully interpreted so as to shoe-horn the data as support for an apposition eye conveniently matching that of a malacostracan crustacean. The authors use their interpretation as a vehicle to criticize several recent papers on Lower Cambrian eyes that likewise claim apposition eyes as the ancestral type. It is unfortunate the authors specifically criticize material, on which those papers are based, when their own material is so very ambiguous.

The paper is sloppily written, possibly in haste. illustrations are of low quality. Certain panels in the figures are not described in the text and the fourth figure is entirely absent.

The reviewer is right and now we corrected the numbering of the figures

The reasoning and logic used to support their data and deride existing publications is often muddled and without factual basis.

What follows is a line-by-line review of the text.

Lines 10-11. "This is in particular true for the eyes of fossils." But the authors then later negate their own opinion by rejecting the interpretation of the radiodontan eye type.

We reformulated the whole issue. The criticised sentence was omitted and there is no contradiction left.

Line 16. "(1h,j)". The standard procedure is to number consecutively starting from the lowest.

We fixed that.

Line 21. ".. "the evidence is equivocal...." not at all: the cited authors present quite respectable evidence, superior to the ambiguous images present in the present manuscript.

We omitted this statement.

Lines 22, 23. "Moreover, the described structures might be in fact cuticular processes as in horseshoe crabs." Not sure what "in fact" is supposed to infer. In any event, this comment is distinctly odd coming from an author who has suggested trilobites are mandibulates.

We reformulated this sentence and omitted this part.

Line 36. Fig. 1 is a mélange of images, not all from trilobites. To what are the authors referring?

This is more specific now.

Line 49 "...as the cones in xiphosurans." Should be written: "as are the cones in xiphosurans."

Done

Lines 49-50. ".....a histological section through the exuvia of a horseshoe crab..." This is not a section but a fracture, and the preservation is so poor that the external features of the lenses appear to have collapsed. There is no reference to Fig. 1k, which so long as it's not mentioned remains irrelevant to the description.

The reviewer is right, we corrected the sentence but by purpose we have chosen an exuvia to show that the cuticular cones are still present and not shed. Of course, exuviae do not look like freshly killed animals but the details that are important are clear in this image.

Lines 76-77. "To test whether the superficial lens-like structures are original or of diagenetic origin, we investigated other cuticular structures of the body of the fossil using SEM." The reasoning is flawed. Just because one area of cuticle surface appears to have a regular structure does not mean that other areas have been spared diagenetic alteration. Likely diagenetic artefacts are nicely indicated at and beneath the surface of the so-called "lenses" in figure 2f.

We think our argument is valid. If the preservation is not just a mould but contains the animal' surface, then the information about cuticular structures is much more detailed. Of course, not all body parts are necessarily equally well preserved, the likeliness is much higher that we deal with a preservation of the cuticle of the eye, if the surface of the cheek and the eye is confluent on the surface and internally as the synchrotron scans reveal.

Line 78. Fig. 2 is a medley of images, none which show head shield cuticle.

The head shield is shown in Fig. 2i.

Line 101. "However, there are cases of unexpected fossil details¹³." What relevance has this paper - which described issues pertaining to soft tissue preservation - to do with the present manuscript?

We think the relevance is of a more general type. Soft tissue preservation is what we have to expect, if cellular crystalline cones and microvilli of a rhabdom are preserved. In any case the soft tissue has been replaced or filled to be still visible.

Lines 104, 105. "Rather the growing calcite crystals replaced the soft internal eye parts and assumed their shape." This is an assertion. But as far as one can see there is no experimental evidence to support it.

The mineralised replacement of tissue has been demonstrated experimentally (e.g. Purnell et al. 2018). Here this is a tentative statement of the kind used in many palaeontological publications. We reformulated the sentence even more tentatively.

Lines 123-124. "Taking all this together, we suggest that there is good evidence that these structures indicate the occurrence of the mandibulate eye type in Trilobita." There is no compelling evidence in this manuscript to support this claim. Authors do themselves no favors by citing other papers on the topic of doing less. Indeed, at least one of the cited papers showed evidence that excludes limulus-type optics in trilobites proposing instead that the ancestral trilobite eye was an apposition type. Have the present authors showed anything more substantive? Note really. Their main piece of

evidence, Fig. 2f, is replete with ambiguity; drawing a lens and cone onto wholly asymmetric structures is no substitute for actual evidence and is a strategy that could expose the authors to ridicule. Likewise, Fig. 2d, which shows columns suggesting visual units, fails to provide any evidence of crystalline cones; the labels "cc" indicate wish-fulfillment by whichever author placed the labels. Returning to Fig. 2f, even if the outline l and cc was a true indication of the dioptric apparatus, which it is certainly not, there would have to be an explanation as to why the claimed lenses (labeled l) are so enormously far apart from each other. There is no evidence of lenses that should be partly visible, albeit offset, between the fractured lenses (if that is what they are) such as would typify the apposition hexagonal mosaic. Nor does the fractured eye of *N. integer* help, as the preservation is so poor that would the figure legend not have stated what it is, there would be no way of knowing it belonged to a compound eye

We omitted the eye of Neomysis integer and replaced it by a surface rendering of the Archegonus lenses and cones. The latter demonstrates the shape of the lenses and the cones. Furthermore, it shows that the cones are morphologically independent structures, since they are missing in some ommatidia.

Lines 125, 126. "Some authors suggested that trilobites possessed crystalline cones without direct evidence." This is true. And judging from the claims made in Fig. 2f one could add the present paper to that list. The Schoeneman et al. 2017 paper on the lower Cambrian *Schmidtiellus reetae* thus remains the most convincing description yet. And yes, the ommas are spaced further apart than in an extant apposition eye, but one must consider the status of *Schmidtiellus* as not simply a Cambrian specimen, but one from about 525 mya, thus coeval with *Radiodonta* and other stem euarthropods.

We reformulated the sentences. Opposite to the reviewer, we think that the radiodontan eyes show that it is unlikely that the Schmidtiellus eye is ancestral in all its structures. Even more, if one accepts the existence of a crystalline cone in radiodontans (see Strausfeld et al. 2016). At least, densely packed hexagonal lenses apparently existed before the trilobites appeared in the tree of life.

lines 134-137. "Our study on eye structures in other trilobites such as *Asaphidae* and *Proetidae* provides additional evidence for the existence of crystalline cones and thus supports the suggestions of Schoeneman et al." That's not very generous is it? Surely it should read "Our study on eye structures in other trilobites confirms the observations of Schoeneman et al." (although, as remarked already, this would be a generous self-appraisal as Figure 2f hardly confirms anything. And the present study is on one trilobite species, so I'm not sure what evidence is shown here for the "other" trilobites.

We reformulated the sentences and hope the relationship between our data and those of Schoenemann et al. are more clearly expressed.

Lines 137-140. "In addition, the cones in the asaphid and *Archegonus warsteinensis* are related to proper cuticular lenses and the eyes comprise more than thousand ommatidia. Furthermore, we present the first evidence for the existence of a rhabdom in a trilobite apposition eye complementing previous findings on retinula cells." 1. What does it matter how many ommatidia (actually lenses) were counted; 2) and, no, it is Schoenemann et al. who provide the first evidence.

The number matters, because the low number of "lenses" was (among other things e.g. the age of the fossil) tentatively interpreted as indicating the primitive status of the Schmidtiellus eye. We think that

there is evidence that this eye is derived under several aspects (page 8, last paragraph). As far as we know, we show the first longitudinal aspect of the rhabdom. Schoenemann et al. show a rosette-like structure. Which is interpreted as cross section through a set of circularly arranged retinula cells. The rhabdom is only inferred.

Lines 142-145. "Moreover, trilobite eyes show a structural diversity that is comparable to that of Recent crustaceans. Like Schmidtiellus reetae the eyes of branchiopod, leptostracan, and amphipod crustaceans possess crystalline cones but lack cuticular lenses." Three taxa lacking cuticular lenses do not Crustacea make! As Dan Nilsson once remarked, the Brachyura alone have more eye types than found across all other crustaceans. So far, four types of trilobite eyes have been described, only two of which consist closely packed facets suggestive of apposition eyes. And why contrast decapods. isopods, stomatopods? These too are recent crustaceans. This reviewer fails to grasp of this

We did not say that the diversity of trilobite eyes is the same as in crustaceans! We are aware of the many different crustacean eyes. We related this sentence just to the fact that some crustaceans possess lenses and others not. But to avoid misunderstandings, we reformulated the sentences.

line 160-166 and 167-170. "Yet, in two recent papers, it was claimed that the compound eyes of the Cambrian arthropod species Fuxianhuia protensa (Fuxianhuiida), Fortiforceps, and Lyrapax unguispinus (Radiodonta) possessed crystalline cones^{15,25}. Since radiodontan species in particular have been considered as stem lineage representative of arthropods ^{2,23}, this would mean that the earliest known faceted eyes were already equipped with the complex dioptric apparatus of modern day mandibulates. For several reasons this is hard to accept:..." It is? Only if the authors offer evidence for an alternative, which they do not. Rather, they take issue with the radiodontan fossil eye, the preservation of which shows a period repeat. This contrasts to the (for want of a more apt descriptor) specious crystalline cone and lens drawn onto what is a highly irregular and asymmetric matrix in their Fig. 2f. Theirs is an example of throwing stones when living in a glasshouse, usually unadvised. While this reviewer has no problem with the authors suggesting a "possible" crystalline cone, what is objectionable is constructing what in Germany is known as a "straw man" to amplify an assertion when it should be a modest suggestion. Taken together, the authors' criticism of Schoeneman et al., (2017) as being less convincing is moot when the present authors', own study appears even less persuasive than it did at first. Nor does the radiodontan description alluded to appear any less persuasive than the images in the present Fig. 2f; indeed, the contrary.

The image of the radiodontan eye in (Strausfeld et al. 2016) is not sufficient to claim the existence of a crystalline cone. Period repeat is just not enough. Cuticular cones as in Xiphosurans are also periodically repeated.

Lines 170-172. "There is no indication that the eyes of Limulus underwent a reduction of a crystalline cone since there are no cells corresponding to cone cells." I suggest the authors go deeper into the literature and consult W. H. Fahrenbach (1975). The visual system of the horseshoe crab, Limulus polyphemus. Int. Rev. Cytol. 41, 285-349, which further suggests that the limulus eye type is derived from an ancestral apposition organization (as would the Leanchoiliid eye).

We do not doubt that the Limulus eye shows an apposition organization. All we want to say is that there is not enough evidence for the claim that Xiphosura lost crystalline cones and replaced them by cuticular cones. Leanchoilia is a good argument to claim the opposite because in several cladistics

analyses it is resolved as stem-lineage euarthropod.

Lines 177-179. "This falsifies hypotheses that xiphosuran eye morphology is related to a 178 certain degree of reduction of an originally more complex situation." Poor Professor Lindström, falsified after 116 years.

This was an error concerning the reference number. We deleted the whole aspect.

Lines 179-183. "Since the data on early crystalline cones in early Cambrian arthropods are elusive, it would be interesting to test whether the compound eyes of Radiodonta, Fuxianhuiida, and other Cambrian stem-lineage arthropods show the xiphosuran eye type or indeed possess crystalline cones. This would clarify the direction of arthropod eye evolution and contribute to the understanding of the evolutionary pathways of these important organ systems." Elusive data; now that is something new this reviewer. However, in the event the lines above may have got lost in translation, what the authors are saying, in none too subtle a manner, is that "We don't believe all that stuff claiming apposition eyes in radiodontans et al., and, dear reader, neither should you." I'd hoped this kind of arrogance had gone extinct around the mid-70s.

As mentioned before, it is not the apposition eye of radiodontans it is the claim that they differentiated crystalline cones. We think that this far reaching claim needs to be more substantiated. Apposition eyes can cuticular cones or crystalline cones (see Land and Nilsson 2012). We specified our criticism of the evidence for the existence of crystalline cones in radiodontans and fuxianhuiids in much more detail now (page 10 f).

Reviewer #3 (Remarks to the Author):

In this contribution, Scholtz et al. describe in great detail the fine structure of trilobite eyes, with particular emphasis on the discovery of reasonably convincing crystalline cones, and make comparisons with a recent study suggesting the presence of these features in admittedly less well-preserved material from the lower Cambrian (Schoenemann et al. 2017, PNAS). The new morphological evidence is clear and this study represents a valuable contribution towards a better understanding of the evolution of optic systems in early arthropods. I have a number of suggestions, however, that include providing a more balanced account of the discussion of these findings, particularly with regard to the phylogenetic implications for trilobites, and some formatting suggestions. However, these are for the most part necessary to make the message even clearer, and thus I will be happy to support publication of this paper in Nature Communications after the authors make what accounts to a moderate revision.

General comments:

- There is a general, and rather troublesome, formatting problem in that material is described in the results, but there is no material and methods section that explains anything regarding the provenance of these fossils. For instance, half of the main data in the paper (e.g. Figure 1) is based on an "unidentified asaphid from the Lindstrom collection, Stockholm". However, there is no mention of age, geographic origin, geological context, stratigraphic provenance, or even a museum catalogue number. All of this is essential data that should be clearly provided in the main text. I realize that

some of this information is provided as supplementary data for some of the figured specimens, but given the brevity of this manuscript I find no reason to not feature it up front in the main text.

We included all the missing information in the first paragraph of the methods-section. .

Figures.

The morphological detail preserved in the fossils is quite impressive; however, the use of small panels induces some frustration given that they show interesting details but their size does not allow a clear examination. Please reformat the figures to avoid the use of small inset panels (e.g. Figure 1c, f, g; Figure 2c) and instead display all the available data as clearly as possible.

We enlarged all small figures.

A second suggestion is the inclusion of a reference diagram that informs the reader about the basic organization and diversity of compound eyes in extant mandibulates and chelicerates, as well as a morphological reconstruction of the eye structure of trilobites (both those reported here, and those of Schoenemann et al. 2017), for clarity. Given that most of the photographs are high magnifications, it is occasionally difficult to identify the structures that are being discussed. A summary and comparative diagram would be extremely helpful to better communicate these points and make the message of the study all the more accessible.

We added a figure with schematic representations of euarthropod eye types and highlighted one optical element of Archegonus with the same colours.

Lines 1 – 31. I get the sense that there should be an abstract here, but this reads more like an introduction to the rest of the paper. I am not aware of Nature Communications have an article format that does not include an abstract, so the authors should include the relevant section here to summarize their findings.

We included an abstract.

Lines 21-23. Please elaborate on how the preserved eye structures in the material of Schoenemann et al. (2017 PNAS) are unusual compared to recent representatives, otherwise this comment comes across as dismissive and not particularly constructive. I realize something on these thread is mentioned in lines 128 – 134, but those do not make it clear why this organization is incomparable with that of extant forms either.

We reformulated the sentences.

Line 29. The authors suggest that the crystalline cone is ancestral within trilobites. Given that their material is of either unknown stratigraphic provenance (the uncertain asaphid), or of upper Devonian age (Archegonus), or correspond to demonstrably derived trilobite groups (e.g. asaphida, proetida). Based on this sparse data sampling, it is impossible to claim that this type of organization is unequivocally ancestral for trilobites unless their findings agree with those of Schoenemann et al. (2017 PNAS), which belong to a stratigraphically and phylogenetically trilobite. Then, in line 139, the authors state that they actually do agree with Schoenemann et al. (2017) after all, which makes me

wonder why they bring up this study earlier as somehow flawed or incorrect in Lines 21 to 23. I recommend to balance the criticism of Schoenemann in the introduction, which gives the erroneous impression to the reader that the data in that paper is somehow fundamentally flawed or unreliable, and that the authors present their findings in the context of the current understanding of trilobite vision more clearly.

We omitted the contradictory statements. Nevertheless we think that based on the current views on trilobite relationships some inference on trilobite eye evolution are possible. Stratigraphically older does not mean that all characters are ancestral. We hope our argumentation is now clearer.

Line 34. Which are the new trilobite finds? This type of lines make it clear why the materials and methods should be part of the main text.

done

Line 44. Please include a reference for a study on modern arthropods here.

done

Line 76. Note that it is entirely possible to have a well-preserved exoskeletal cuticle but diagenetically altered eyes in fossil arthropods (e.g. see Goggllops ensifer in Siveter et al. 2017, Geological Magazine). Also, what specimens are we talking about here? Based on the following lines I imaging it is Archegonus, but if that is the case, this is conveyed rather awkwardly. Please reorganize this section for clarity.

We reorganised this section and hope it is now clearer. We know that different body parts can show a different degree of preservation, but in this case there is a continuous surface between the eye and the other parts of the head of Asaphus sp. (see Fig. 2j).

Lines 80 to 105. This descriptive section further makes a point for including an additional figure with diagrams that guide the reader through the preserved morphology, as it is difficult to distinguish some of the features mentioned in the main text (e.g. clear zone in Figure 2e).

We added some labelling to the figures indicating the clear zone.

Also, does the CT data provide further evidence of the organization of the lenses in the eyes of Archegonus? Figure 2f is a bit too suggestive as there is not a perfectly clear-cut morphological distinction between the lens and the crystalline cone on the exposed surface. Is this could be demonstrated with the CT data obtained it may be possible to segment and isolate these elements to obtain a better view of their morphology.

We added a surface rendering image of the scan showing the morphology of the lenses and the separate crystalline cones (Fig. 3f).

Line 143. The authors mention that trilobite eyes show a structural diversity comparable to that of recent crustaceans. However, it is not particularly clear that they have described structurally different morphologies based on the unnamed asaphid and Archegonus. Is this diversity based only

on the length of the crystalline cone? If so, again, a diagrammatic reconstruction would help to better convey these nuances, as they are not evidently clear from the figures alone without a side-by-side comparison.

Of course we are aware of the fact that the general diversity of crustacean eyes is much greater than that of trilobite eyes. The comparable aspects of diversity relates to lens forms, the distances between the ommatidia, and the shape of crystalline cones between Schmidtiellus and the two species studied by us. We think this does not require an additional figure. Accordingly, we tuned down the comparison with crustaceans and reformulated the whole section (page 8).

Line 165. It is true that the phylogenetic position of trilobites is still debated, which is probably due to their proximity to the main dichotomy between chelicerates and mandibulates. Given these considerations, the presence of mandibulate-like ocular structures in trilobites can equally be interpreted as symplesiomorphic relative to crown-group Euarthropoda, in line with recent suggestions that the brain outline of Fuxianhuia (Ma et al. 2012, Nature; Ma et al. 2015, Current Biology) and some Cambrian bivalved forms (e.g. Odaraia alata, see Ortega-Hernandez 2015, Current Biology), has a mandibulate-like morphology. In this context, it is just as likely that the ocular organization of trilobites, as well as other mandibulate-like features, are actually ancestral rather than synapomorphic. I understand if the authors object to his alternative, but they should at least acknowledge that this scenario is equally supported by the available data, rather than focussing the discussion on their preferred hypothesis.

We reformulated this part of the discussion and we think it is more balanced now.

Line 166. Again, to mention that the "... presentation of the data is not convincing..." is a somewhat hostile but also superficial criticism. Please explain how it is that this data is unconvincing, rather than just claiming it so. For example, I would agree that the preservation of the eyes in Fuxianhuia protensa do not allow to recognize extremely fine details of the internal eye organization, such as the presence of crystalline cones, as demonstrated in Figure 3f of Ma et al. 2012, Nature.

We added a sentence about preservation (page 10, 2nd sentence) and we are much more specific in our criticism of the figures on Radiodonta and Fuxianhuia eyes page 10, 3rd sentence ff.). Furthermore, we added an own picture of an eye of Fuxianhuia (Suppl 1.).

Line 177. Following from the former comment, the authors mention that the eye structure of eurypterids demonstrates that there has been no loss or reduction of the crystalline cone in Xiphosura. However, none of the references cited provide information on the internal organization of the eyes in eurypterids, and as far as I am aware this information is not directly available from fossils, but instead focus on the superficial arrangement of eurypterid eyes and how that informs their ecology. Thus, there is no such demonstration that the eye organization of xiphosurans cannot possibly be secondarily derived. In this context, it is noteworthy that lateral eyes have been gained and lost repeatedly throughout xiphosuran evolution (e.g. see phylogeny in Lamsdell 2016, Paleontology), and thus it is entirely expected that these will represent highly plastic structures. The claim that the eyes of extant xiphosurans reveal the ancestral condition of arthropods is clearly focused on extant diversity alone, and thus probably best discussed with a pinch of salt.

The information about the inner organisation of eurypterid eyes stem from Clarke and Ruedemann

and Wills. The other references are only used to show that this eye type can comprise numerous lenses and ommatidia. Lamsdell's 2016 paper does not show or discuss a frequent loss and regain of compound eyes in xiphosurans. Nevertheless, we deleted the entire section on eurypterids.

Reviewers' Comments:

Reviewer #1:

Remarks to the Author:

I think I must have read the earlier version of the paper with an eye worker hat on rather than a trilobite worker hat because I missed a glaring error that, thankfully, leaped out on re-reading.

One of the two trilobites studied in the paper is a putative *Asaphus* sp., from the Silurian of Gotland. *Asaphus* is an Ordovician genus, and the entire family Asaphidae went extinct at the Ordovician-Silurian boundary. There is absolutely no chance that there is an *Asaphus* from the Silurian anywhere. The rocks in outcrop on Gotland are a layer cake sequence through most of the Silurian, from the Upper Llandovery through the Upper Ludlow. There is some Upper Ordovician in the subsurface. Even if this specimen were *Asaphus* it would be from the Ordovician rather than the Silurian. I suspect it is actually Silurian (that being the geological period from which the beautiful trilobites of Gotland derive) and it is misidentified. It is essential that the authors figure out what this taxon actually is.

Some rhetorical sentences are added on lines 271-275. The authors don't mask the fact they want to stick the boot into Strausfeld et al., but casting these sentences in the interrogative is kind of tacky.

Reviewer #2:

Remarks to the Author:

Review of NCOMMS-18-10188A

This revised paper amplifies many of the weaknesses of its first version and further embroiders a tangential discussion recommended in the first review for stringent shortening if not complete deletion.

The authors attempt to persuade the reader that they have identified evidence for trilobites possessing apposition retinas, therefore conferring their status as Mandibulata (currently, Myriapoda+ Pancrustacea) and demonstrating that this type of retina was ancestral to Mandibulata. The claim minimizes other papers that have previously proposed Trilobites as mandibulates mainly with reference to their appendicular traits, with one recent account demonstrating the mandibulate nature of the second postantennular appendage pair (Zeng et al., *Geology Magazine*, 156, 1306–1328). Evidence from another lower Cambrian trilobite, published in PNAS (their reference 19) makes a very strong case for apposition optics. Indeed, the case is stronger because that specimen is the earliest eye known from the fossil record (approx. 520 mya) whereas the species studied by Scholtz et al. are relatively recent (approx. 485-358 mya). At most, the present account by Scholtz et al., would be confirmatory. If what the authors present had made a compelling case for apposition optics this would show that this eye type had been retained relatively late into Trilobite history.

The authors claim images of a fossilized rhabdomeric structure. The demonstration of fossilized rhabdomeres alone does not provide any information about the optical properties of the eye, and to demonstrate a tetraconate apposition eye, it is necessary to provide convincing evidence of a cruciform organization of the crystalline cone in a Trilobite. Although the authors show numerous images that they think may have been cones in a Silurian Trilobite, these are inference-based assumptions, not data-based. Their one specimen used to claim fossilized crystalline cones, criticized earlier for having outlines imposed on the photograph to indicate assumed corneal boundaries, is now presented with overlaid coloring. The colors obscure the structure beneath. Other parts of the image are of insufficient clarity to provide evidence for discrete components of an apposition eye. The comparison of dimensions pertaining to the exposed retinal depth with those indicated on the volumetric scan of the same area accentuate rather than clarify ambiguities.

The authors are not the first to think they have provided conclusive evidence for apposition eye ancestry. Yet, working from a relatively recent Trilobite species can't provide a proxy for an ancestral morphology. Perhaps recognizing the fragility of the data, it was felt appropriate to focus much of the discussion on a paper on early eye published elsewhere by another group of researchers. This is a distraction, here compounded by misinterpreting that paper's findings.

The following details what I see as serious defects in this work.

Line 18, 19. The literature on *Asaphus* sp. ascribes this species to the upper or middle Ordovician, not the Silurian. The literature on (*Waribole*) *warsteinensis* assigns this species to the Lower Carboniferous.

Line 27. Euarthropods?

Lines 29-31. "Hence, there is an enormous body of literature dealing with developmental, morphological, physiological, and evolutionary aspects of this important sense organ¹." Reference 1 is a text book dealing with animal eyes in general. The statement should cite at least one or two reviews that deal with the value of compound eyes in phylogenetic analyses, for this what the authors propose in their first sentence of this manuscript. I think many would say that compound eyes have less phylo-indicative value than what these authors assume.

Lines 33-34. "...those that have a cuticular lens with a cone-like extension, fulfilling a similar purpose of collecting light and guiding it to the retinula cells^{6,7}." Reference 7 is a description of the nature of the *Limulus* cuticular cones, which Exner (reference 6) used for the first description of apposition optics. Reference 7 is the first of a series of paper by Fahrenbach on the *Limulus* eye. It introduces the possibility of cells homologous to Semper cells, which in Mandibulates secrete the quadripartite crystalline cone. That the present authors cite this first Fahrenbach paper to advocate that *Limulus* has no equivalents of crystalline cone cells suggests that they haven't read that account closely. Fahrenbach's second paper (its citation still omitted, ignoring the recommendation in the first review to include it) demonstrates that *Limulus* possesses attenuated cone cells (a crucial line of evidence supporting the *Limulus* type eye is likely derived from an ancestral apposition retina).

Lines 35, 36. "...often been interpreted as an ancestral character for euarthropods^{8,9}." Two references do not make this "often" especially as neither provides compelling evidence that the *Limulus* type cuticular apposition eye is ancestral for euarthropods.

Lines 37, 38. "...specialized cone cells occurs in some myriapods, crustaceans, and hexapods and has been interpreted as an apomorphy of Mandibulata^{10,11}." This is true, but the citations select the exceptional occurrence of cone cells in basal Diplopoda and Chilopoda, thus demonstrating that in most Myriapod species eyes depart from typical apposition cone type eyes. (Considering the derived Schizochroal type eyes of Trilobites, the authors might have made more out such possible divergent trends). I would have thought that Richter's now classic 2002 paper (in *Organisms Diversity & Evolution*) would be the one to cite at this juncture.

Line 39. "The origin of compound eyes dates back at least to the Lower Cambrian" this needs at least one citation: perhaps reference 14.

Lines 41, 42. "Most details about fossil compound eye structures stem from investigations on trilobites¹⁵⁻¹⁷" This statement contradicts the next sentence: that previous work on Trilobite eyes

was only on the cuticular structure (external morphology) of eyes. Albeit interesting, those studies cannot be described as "detailed." In fact, there are a number of papers on eye structure of Lower Cambrian taxa, not least those by Schoenemann and colleagues on Chengjiang and earlier specimens. The authors cite the latter (reference 19) but in this revision seem as resistant to its implications as they were in their original account.

Lines 56-63. Three remarks.

1. "...fossil preservation of a longitudinal section through a rhabdom indicating that early arthropod compound eyes possessed the same type of light receptors as those of some modern arthropods." But the extant species for comparison has a superposition eye, not an apposition eye, so it does not offer proper comparison with a trilobite retinula if that retinula belonged to an apposition eye.
2. "We suggest that a crystalline cone is ancestral within trilobites." Even if shown, it could be convergent. The species discussed are from relatively late in Trilobite history.
3. "our results may corroborate the proposed close phylogenetic relationship of trilobites to Mandibulata." This relationship has been proposed several times, with emphasis on the gnathal morphology of the second postantennular appendages.

Results. To spare repetition I will refer to the figures and legends.

Fig. 2. Assuming that the hexagonal arrangement of facets in panel b is offered as support for the eye being an apposition eye (otherwise, why show this image?) then it is as justified to show surface views of facets (at far better resolution) of the radiodontan eye (see Patterson et al.) to suggest the same. Of significance is that regular hexagonally profiled facets are indicative of apposition eyes (see early papers by Land) whereas facets of *Limulus* and Eurypterans show irregularities. Panel c: the "cut" section does not indicate the level or the orientation, and this should be indicated in panel b. The "holes" in panel c do not show crystalline cones, they show holes. Panel d: this does not show cones but shows electron translucent areas that the authors assume were cones. But tetraconate cones reveal membrane borders demarcating their constituent cell borders. These are not shown here. Panel f: there are no obvious lenses in this figure, and the layer beneath the one indicating lenticular cuticle does not show cones but shows "holes" claimed as the absent cones: thus, an assumption, not a demonstration. Assuming an absence does not constitute evidence of past presence. Panel g: assumptions are made that blurry profiles beneath a blurry surface must be cones. This is as unconvincing as the "artefactual cone layer" shown from Fuxianhuia in suppl. Fig. 1. Panel h: a section of the eye of the centipede *Scutigera* purports to show structures indicated in panel g. But the absence of any clear correspondence is compounded by the lack of a focused image in g. Panel i does not provide any contribution to the cone question. Panel j: This image shows the eye surface (which is not indicated by the authors). Beneath the layer of lenses (here referred to as cuticle) arrows indicate absent cones. However, it is an act of faith that what are absent were once present. Panel k: this shows the assumed sloughing (exudial) of the trilobite cuticle. However, the panel provides no supporting evidence for cones.

Fig. 3. Panel a: The image has been taken at such low magnification that the transition of eye

to head is not apparent. The asterisk may possibly indicate a basal membrane, but it could equally be a smooth fracture. If preservation was good enough to maintain features of photoreceptors (which are neurons) as claimed, then one would expect to see preserved retinula axons (these are neuronal prolongations of photoreceptors) extending further beneath the basement membrane. There is no evidence for this here. Panel b inset: what is meant by a putative optical unit? The image suggests the inner faces of the cuticular lenses. It is unfortunate that the resolution is poor and the magnification low. Would it be sharper and at higher magnification, then if the "optical units" would reveal the inner face of the dioptric apparatus the observer might be able to see and count the tips of photoreceptors abutting the crystalline cone which is preserved should show its cruciform composition. Panel c; Here the fractured eye shows longitudinal subdivisions, the periodicity of which approximately corresponds to surface structures interpreted as lenses. However, arbitrarily labeling a level beneath the surface as crystalline cones may, to the authors, be where they expect to see cones but for the reader there is nothing to suggest a differentiated structure at that level. The image of crystalline cones of the extant eumalacostracan krill *M. norvegica* in panel d is no help in that it amplifies the absence of a corresponding structure in panel c. Panels e, f, and g: these are the images pivotal to the authors' claim of a trilobite apposition eye. However, using the scale bars to compare the putative cones in the exposed eye depth (panel e) with the surface rendering (panel f) one arrives at a putative cone depth (the green area painted on panel f) of about 50 microns in e whereas the structure indicated as a cone in panel f would be less than half that. The image of the rhabdomeres in panel g, from the extant *M. norvegica*, is somewhat misleading. These photoreceptors are far removed from the dioptric layer due to the intervening clear zone (filled with column like structures, which elicit some awkward but obvious questions) of an eye that is an optical superposition eye, not an apposition eye. If the authors are claiming that the trilobite eye was an apposition eye why haven't they used a crustacean proxy with an apposition eye? Never mind. What is a most reasonable interpretation of a rhabdomeric structure in panel e appears to correspond to the layered arrangement of rhabdomeres in *M. norvegica*, However, despite the colors ascribed to what the authors assume are elements of an apposition retina, the claim that the dioptric apparatus of *A. warsteinensis* comprises a cuticular lens succeeded by a crystalline cone is not defined well enough to be convincing. The authors may have convinced themselves, but this reviewer cannot see any indication of a separate tetraconate crystalline cone disposed beneath, and distinct from, a clearly resolve inner face of a cuticular lens; nor, proximally, a crystalline cone's delineation from the apices of underlying photoreceptors. Such features would be required for differentiating a crystalline cone from deeply penetrating cuticular lenses.

Lines 153-156. "In particular, the preservation of the internal eye anatomy of the Jurassic thylacocephalan crustacean *Dollocaris ingens* reveals many details including crystalline cones and a rhabdom structure that resemble the pattern found in the trilobites studied here²⁰." This comment is an extraordinary distortion: if anything, the converse applies. The exquisite preservation of the dioptric apparatus elements and rhabdoms of *D. ingens* sets the benchmark for any comparison. In Vannier et al.s' paper each element is unambiguously identified and delineated. It does not resemble "the pattern in the trilobites studied here" because there is no definitive pattern and only one trilobite species reveals anything that might approach a

rhabdomeric structure comparable to those illustrated from *D. ingens*.

The Discussion.

In my review of the original manuscript I pointed out that spending a lot of type criticizing other studies to shed a kinder light on one's own work was a strategy that can easily backfire. Yet instead of taking this advice to heart, the authors have further embroidered and extended an already lengthy polemic levelled against earlier studies of Cambrian eye morphologies. While Scholtz et al. are entitled to critical opinions, if these are submitted for publication then they have to stand up to scrutiny: are they accurate; are they informative; do they strengthen their own observations? The following selections illustrate that none of these criteria pertain.

From lines 184-210, the Discussion reads as a sober assessments of trilobite eye studies concluding with the author assertion that their study supports the conclusion of that of Schoenemann et al, in PNS 2017. In short, the data of the present account, if correct, would confirm Schoenemann et al's claim that the trilobite eye was an apposition eye type and because Schoenemann's species derives from the lower Cambrian (even deeper than the Chengjiang biota) it is likely to represent the earliest apposition retina.

From line 211 things begin to get decidedly strange, beginning with an obscure argument that some minor oddities of Schoenemann's early Cambrian specimen, such as widely separated facets, suggest that its apposition characters are evolutionarily derived. But derived from what? The reader is left uninformed. And if they were derived, why couldn't the trilobite eyes claimed in this paper as apposition type eyes likewise be derived?

But then the authors make the serious mistake, when discussing work on Chengjiang fossil eyes, in questioning why one species of "Megacheira" possess Limulus like cuticular cones whereas another species possessed crystalline cones for apposition optics. I will return to this point presently. The authors go on to ratchet up their argument by challenging fossil preservation, the quality of the specimens, and more besides. I have selected just a few of these comments, not because they have anything to do with the matter at hand – namely trilobite eyes – but as a demonstration of why it is wiser to think carefully before setting up a straw man scenario. It's a strategy that can seriously backfire, as it has here.

Lines 255-259. Scholtz et al write: "First of all it is astonishing that extremely detailed structures such as crystalline cones, which are made up of just a few cells, should be preserved despite the complex fossilization process of the Chengjiang Lagerstätte that involves flattening, microbial degradation, chemical alteration and replacement³⁸." This rebuke ignores the established fact that the trilobite has a calcareous exoskeleton that resists flattening, is impermeable and is therefore notorious for precluding internal fossilization. That being established, isn't it the more "astonishing" that cone cells and rhabdomeres protected by a calcareous exoskeleton would nevertheless be fossilized? Despite the authors' astonishment, they know that there is nothing astonishing about preservational processes providing soft tissue preservation in soft cuticle Chengjiang species. They cite Parry et al. (BioEssays 40, 2018) as a

source reference for exceptional preservation. They have likely read Ma et al., 2014 describing taphonomic events providing exceptional preservation of *Fuxianhuia*, for example. However, Scholtz et al. underpin their “astonishment” by citing a paper by Liu et al 2018 that is devoid of any insights into the process of preservation but which instead trivially conflates the identification of organs with biofilms, the asymmetries of which disqualify their interpretation of soft tissue.

Lines 273-275. The authors ask, “Why, for instance, did they (the authors of reference 3) interpret the cone-like structures found in the eye of *Fortiforceps foliosa* as crystalline cones and those in *Leancoilia illecebrosa* as cuticular cones?” That the authors ask this question is perplexing as there are two clades comprising fossils traditionally accorded to “Megacheira.” “Megacheira,” sometimes referred to as great appendage (GA) arthropods, is not only exemplified by species such as *Leancoilia illecebrosa* and *Allalcomenaeus* where two pairs of eyes are set flush with the cuticle: it is this 2+2 eye type that possess cuticular type cones. In contrast, *Fortiforceps*, also historically grouped in “Megacheira,” is a member of a clade multisegmented GA arthropods. This second clade comprises species that lack the discrete tagmosis of *Leancoiliids* and which, despite having stubby pincer-like deutocerebral appendages, have just one pair of eyes, each eye surmounting an eye stalk, as do other multisegmented “Megacheira” e.g. *Pseudoiulia cambriensis* and *Jianfengia multisegmentalis*. There are obvious morphological distinctions of eye types belonging to GA arthropods (with two distinct morphologies of GAs) that differentiate two distinct clades of “Megacheir” and the morphology of their deutocerebral appendages further differentiate Megacheira: those with double eyes possess long, delicate caliber-like “great appendages” whereas multisegmented Megacheira, with single pairs of compound eyes, possess short pincer-like GAs, as mentioned above. It is important that criticisms of published works do not suffer this sort of confusion.

Lines 278. “...the phylogenetic position of megacheirans and fuxianhuiids is still controversial.” The authors omit any citation in support of this statement, when the most recent phylogenies, such as Legg, D. *A.Nat.Comm.* 4, 2485 (2013), are available.

281 -284. “For instance, if indeed the megacheiran *Leancoilia illecebrosa* possesses a chelicerate eye-type and if it is a stem lineage euarthropod then this renders the structure of the ommatidia of the xiphosuran eye plesiomorphic for crown-group euarthropods.” Yes, as does the eye of *Radiodonta*, and one has no problem with that. However, this has no bearing on their thesis that retinas of two trilobites, one Ordovician and another Silurian, are representatives of a plesiomorphic condition of Euarthropoda that presumably originated in the lower Cambrian or before. Even if convincing, which the manuscript paper is not, descriptions of eyes of late evolving trilobite species can’t serve as proxies for an ancestral morph. The Schoenemann et al. account does, however, being based on a specimen from around 520mya.

lines 285-287 “.....there is no indication that the eyes of *Limulus* underwent loss of a crystalline cone, since there are no cells corresponding to cone cells....” Here the authors are referring to a paper by Fahrenbach (reference 7) but not cited at this juncture. Reference 7 discusses Semper cell-like cone cells. In my first review, I drew attention to the second Fahrenbach paper,

which is here totally ignored, but which describes *Limulus* cone cells. Let me cite the 1969 paper again: "Fahrenbach, W. H. 1969. The morphology of the eyes of *Limulus* II. Ommatidia of the compound eye. *Z. Zellforsch* 93. 451-483). That cones cells are present in *Limulus* negates Scholtz et al.'s contention that the *Limulus* type eye (or the eyes of *Alalcomenaeus* or *Leandroilia*) cannot have derived from a plesiomorphic apposition ground pattern established in deep time.

It is unfortunate that the supplemental figure 1 has been used to argue that inaccurate interpretations have, in the past, been published relating to the eye morphology in the species *Fuxianhuia*. The specimen in suppl. Fig. 1, also *Fuxianhuia*, is illuminated very obliquely with much back-scatter allowing great latitude of interpretations. It appears that the specimen is from a private collection, lacking an accession number which would need to be obtained if the intention is to publish it in a journal.

Reviewer #3:

Remarks to the Author:

Scholtz et al. have addressed all my previous concerns satisfactorily, the manuscript now reads much better, and the figures are simply impressive. Different lines of evidence and argumentation are treated in a much more balanced manner, and even if I maintain a small degree of disagreement over some very specific interpretations or references, I think this manuscript is in good shape for publication.

The only minor, but recurring, issue that needs to be addressed is the ongoing malpractice of using "arthropod" and "euarthropod" as interchangeable terms. I have discussed this issue extensively (see reference below), and whether the authors decide to use euarthropod (more accurate in my opinion) or arthropod (more familiar, but problematic) is up to them, but they should maintain consistency throughout their manuscript and figures. I may also add that using the suggested convention of lower-stem Euarthropoda and upper-stem Euarthropoda from that contribution would help not only with making their Figure 4 clearer, but also in places in the manuscript where radiodontans and fuxianhuids are discussed in the context of the stem lineage.

Ortega-Hernández, J., 2016. Making sense of 'lower' and 'upper' stem-group Euarthropoda, with comments on the strict use of the name Arthropoda von Siebold, 1848. *Biological Reviews*, 91(1), pp.255-273.

Javier Ortega-Hernández (16th Sept 2018)

NCOMMS-18-10188A

Response to the comments of the reviewers.

Reviewers' comments:

Reviewer #1 (Remarks to the Author):

I think I must have read the earlier version of the paper with an eye worker hat on rather than a trilobite worker hat because I missed a glaring error that, thankfully, leaped out on re-reading.

One of the two trilobites studied in the paper is a putative *Asaphus* sp., from the Silurian of Gotland. *Asaphus* is an Ordovician genus, and the entire family Asaphidae went extinct at the Ordovician-Silurian boundary. There is absolutely no chance that there is an *Asaphus* from the Silurian anywhere. The rocks in outcrop on Gotland are a layer cake sequence through most of the Silurian, from the Upper Llandovery through the Upper Ludlow. There is some Upper Ordovician in the subsurface. Even if this specimen were *Asaphus* it would be from the Ordovician rather than the Silurian. I suspect it is actually Silurian (that being the geological period from which the beautiful trilobites of Gotland derive) and it is misidentified. It is essential that the authors figure out what this taxon actually is.

The reviewer is correct and we clarified the issue in the Material and Methods section. Based on what can be seen from the part that we have, it can be pretty securely identified as an asaphid. It is definitely not one of the Silurian species of Gotland. Hence, it must be from the Ordovician. Naturally, it is difficult to reconstruct where exactly Lindström had collected it, but since Gotska Sandön shows some subsurface Ordovician, it is possible that he found an Ordovician species. The Riksmuseet lists the specimen as Silurian and we naively adopted this. But at the time of Lindström, the term Silurian included what we now call Ordovician.

Some rhetorical sentences are added on lines 271-275. The authors don't mask the fact they want to stick the boot into Strausfeld et al., but casting these sentences in the interrogative is kind of tacky.

Yes, we have a critical view on the Strausfeld et al. paper. In the first version of our manuscript we did not want to go into great detail why we do not buy the arguments of this publication, but reviewers 1 and 3 asked for more details. Hence, we added these to the second version. We neither wanted to stick a boot into someone's article nor was it our intention to be tacky, but it must be possible in a scientific discourse to criticize other people's work.

Nevertheless, we deleted the passages criticised by the reviewer and hopefully we found an acceptable form.

Reviewer #3 (Remarks to the Author):

Scholtz et al. have addressed all my previous concerns satisfactorily, the manuscript now reads much better, and the figures are simply impressive. Different lines of evidence and argumentation are treated in a much more balanced manner, and even if I maintain a small degree of disagreement over some very specific interpretations or references, I think this manuscript is in good shape for publication.

The only minor, but recurring, issue that needs to be addressed is the ongoing malpractice of using "arthropod" and "euarthropod" as interchangeable terms. I have discussed this issue extensively (see reference below), and whether the authors decide to use euarthropod (more accurate in my opinion) or arthropod (more familiar, but problematic) is up to them, but they should maintain consistency throughout their manuscript and figures. I may also add that using the suggested convention of lower-stem Euarthropoda and upper-stem Euarthropoda from that contribution would help not only with making their Figure 4 clearer, but also in places in the manuscript where radiodontans and fuxianhuids are discussed in the context of the stem lineage.

The reviewer is right and in the new version we use euarthropods consistently. Thanks for the hint. We added the proposed reference of Hernández-Ortega 2016.

Reviewer #2 (Remarks to the Author):

This revised paper amplifies many of the weaknesses of its first version and further embroiders a tangential discussion recommended in the first review for stringent shortening if not complete deletion.

We deleted the larges part of the criticism.

The authors attempt to persuade the reader that they have identified evidence for trilobites possessing apposition retinas, therefore conferring their status as Mandibulata (currently, Myriapoda+ Pancrustacea) and demonstrating that this type of retina was ancestral to Mandibulata. The claim minimizes other papers that have previously proposed Trilobites as mandibulates mainly with reference to their appendicular traits, with one recent account demonstrating the mandibulate nature of the second postantennular appendage pair (Zeng et al., Geology Magazine, 156, 1306–1328). Evidence from another lower Cambrian trilobite, published in PNAS (their reference 19) makes a very strong case for apposition optics. Indeed, the case is stronger because that specimen is the earliest eye known from the fossil record (approx. 520 mya) whereas the species studied by Scholtz et al. are relatively recent (approx. 485-358 mya). At most, the present account by Scholtz et al., would be confirmatory. If what the authors present had made a compelling case for apposition optics this would show that this eye type had been retained relatively late into Trilobite history.

We did not want to minimize other papers pointing in the same direction and we added the Zeng et al. article to our references, which we simply had overlooked.

There is a huge discussion about the relationship between stratigraphy and character polarity. It is simply not the case that older fossils always show ancestral character stages. We put a sentence and a citation into the text to stress this.

The authors claim images of a fossilized rhabdomeric structure. The demonstration of fossilized rhabdomeres alone does not provide any information about the optical properties of the eye, and to demonstrate a tetraconate apposition eye, it is necessary to provide convincing evidence of a cruciform organization of the crystalline cone in a Trilobite. Although the authors show numerous images that they think may have been cones in a Silurian Trilobite, these are inference-based assumptions, not data-based. Their one specimen used to claim fossilized crystalline cones, criticized earlier for having outlines imposed on the photograph to indicate assumed corneal boundaries, is now presented with overlaid coloring. The colors obscure the structure beneath. Other parts of the image are of insufficient clarity to provide evidence for discrete components of an apposition eye. The comparison of dimensions pertaining to the exposed retinal depth with those indicated on the volumetric scan of the same area accentuate rather than clarify ambiguities.

It would be perfect to see a preservation of the four parts of the crystalline cones to make an even stronger argument. However, this structure has not been preserved in any fossil crystalline cones so far. Nevertheless, the reviewer has apparently no problems with the interpretations of other authors, which also did not show the four cone cells.

We do not see the point of data versus inference. Data have to be interpreted. Even more, we were the first to use the process of elimination to show that these cones are independent from the cuticle

We have chosen transparent colors that do not obscure the structure.

The authors are not the first to think they have provided conclusive evidence for apposition eye ancestry. Yet, working from a relatively recent Trilobite species can't provide a proxy for an ancestral morphology. Perhaps recognizing the fragility of the data, it was felt appropriate to focus much of the discussion on a paper on early eye published elsewhere by another group of researchers. This is a distraction, here compounded by misinterpreting that paper's findings.

We did not claim that we were the first to provide evidence for apposition eye ancestry and this is not a major focus of the manuscript.

Line 18, 19. The literature on *Asaphus* sp. ascribes this species to the upper or middle Ordovician, not the Silurian. The literature on (*Waribole*) *warsteinensis* assigns this species to the Lower Carboniferous.

Asaphus has been corrected to Ordovician. In the more recent literature, Warinbole is clearly Devonian (e.g. Clausen et al. 1989 Spalten und ihre Füllungen in den Carbonatgesteinen des Warsteiner Raumes (nordöstliches Rheinisches Schiefergebirge). Fortschr. Geol. Rheinl. U. Westf. 35, 309-391).

Line 27. Euarthropods?

corrected

Lines 29-31. "Hence, there is an enormous body of literature dealing with developmental, morphological, physiological, and evolutionary aspects of this important sense organ1." Reference 1 is a text book dealing with animal eyes in general. The statement should cite at least one or two reviews that deal with the value of compound eyes in phylogenetic analyses, for this what the authors propose in their first sentence of this manuscript. I think many would say that compound eyes have less phylo-indicative value than what these authors assume.

Lines 33-34. "...those that have a cuticular lens with a cone-like extension, fulfilling a similar purpose of collecting light and guiding it to the retinula cells6,7." Reference 7 is a description of the nature of the *Limulus* cuticular cones, which Exner (reference 6) used for the first description of apposition optics. Reference 7 is the first of a series of paper by Fahrenbach on the *Limulus* eye. It introduces the possibility of cells homologous to Semper cells, which in Mandibulates secrete the quadripartite crystalline cone. That the present authors cite this first Fahrenbach paper to advocate that *Limulus* has no equivalents of crystalline cone cells suggests that they haven't read that account closely. Fahrenbach's second paper (its citation still omitted, ignoring the recommendation in the first review to include it) demonstrates that *Limulus* possesses attenuated cone cells (a crucial line of evidence supporting the *Limulus* type eye is likely derived from an ancestral apposition retina).

Its common practice to cite textbooks if a huge literature has to be covered. The chapter on arthropod eyes in the cited book is long and extremely detailed. It furthermore provides the relevant literature. Moreover, we cite other papers here and throughout the manuscript showing the justification of the sentence. Fahrenbach describes the eyes of Limulus in several papers. We cite only one. The reviewer prefers another one. Even if Fahrenbach considers the transparent cells in Limulus as homologs to mandibulate semper cells, this does not mean that we have to follow his argument. As we write, the number and the specific structure of these cells do not justify homologization to crystalline cones. Limulus eyes act as apposition eyes. We do not question this at all.

Lines 35, 36. "...often been interpreted as an ancestral character for euarthropods 8,9 ." Two references do not make this "often" especially as neither provides compelling evidence that the *Limulus* type cuticular apposition eye is ancestral for euarthropods.

We added some references and altered the sentence a bit.

Lines 37, 38. "...specialized cone cells occurs in some myriapods, crustaceans, and hexapods and has been interpreted as an apomorphy of Mandibulata 10,11." This is true, but the citations select the exceptional occurrence of cone cells in basal Diplopoda and Chilopoda, thus demonstrating that in most Myriapod species eyes depart from typical apposition cone type eyes. (Considering the derived Schizochroal type eyes of Trilobites, the authors might have made more out such possible divergent trends).

The cited articles discuss the other eyes occurring among myriapods, There is no need to repeat this here

I would have thought that Richter's now classic 2002 paper (in *Organisms Diversity & Evolution*) would be the one to cite at this juncture.

Richter is co-author of the later paper Müller et al. 2003 that updated his 2002 article. Hence, we cite the newer one.

Line 39. "The origin of compound eyes dates back at least to the Lower Cambrian" this needs at least one citation: perhaps reference 14.

At the end of the sentence we refer to four papers, among them reference 14.

Lines 41, 42. "Most details about fossil compound eye structures stem from investigations on trilobites 15-17" This statement contradicts the next sentence: that previous work on Trilobite eyes

was only on the cuticular structure (external morphology) of eyes. Albeit interesting, those studies cannot be described as "detailed." In fact, there are a number of papers on eye structure

of Lower Cambrian taxa, not least those by Schoenemann and colleagues on Chengjiang and earlier specimens. The authors cite the latter (reference 19) but in this revision seem as resistant to its implications as they were in their original account.

We do not see the contradiction. Still most papers on fossil compound eyes deal with those of trilobites describing the different lens types and physical properties and inferring evolutionary change. Furthermore, we cite more than one of the Schoenemann papers.

Lines 56-63. Three remarks.

1. "...fossil preservation of a longitudinal section through a rhabdom indicating that early arthropod compound eyes possessed the same type of light receptors as those of some modern arthropods." But the extant species for comparison has a superposition eye, not an apposition eye, so it does not offer proper comparison with a trilobite retinula if that retinula belonged to an apposition eye.
2. "We suggest that a crystalline cone is ancestral within trilobites." Even if shown, it could be convergent. The species discussed are from relatively late in Trilobite history.
3. "our results may corroborate the proposed close phylogenetic relationship of trilobites to Mandibulata." This relationship has been proposed several times, with emphasis on the gnathal morphology of the second postantennular appendages.

Ad 1) Indeed, we show a recent superposition eye. This is meant to demonstrate the similarity of rhabdom microvilli (which are not really different between the apposition and superposition eye types, hence it deals with the identification of preserved microvilli as such) and the difference between apposition and superposition eyes.

Ad 2) This is no real criticism but a thought-terminating cliché. Likewise the eye structures of Schmidtiellus or Fuxianhuia or any other euarthropod could be convergent. As is mentioned above, the age of a fossil is not a necessary criterion to claim a plesiomorphic character state, neither is the younger age an indication of convergence.

Ad 3) Indeed, we never claimed that this is the first account to propose this relationship. However, during the last years this view has been overturned by a number of phylogenetic analyses. Hence we think that additional characters based on eye structure may help to clarify the issue.

Fig. 2. Assuming that the hexagonal arrangement of facets in panel b is offered as support for the eye being an apposition eye (otherwise, why show this image?) then it is as justified to show surface views of facets (at far better resolution) of the radiodontan eye (see Patterson et al.) to suggest the same. Of significance is that regular hexagonally profiled facets are indicative of apposition eyes (see early papers by Land) whereas facets of *Limulus* and Eurypterans show irregularities.

This panel shows the eye of the investigated trilobite species to demonstrate the regular hexagonal arrangement – exactly supporting the argument of the reviewer.

The Panel c: the “cut” section does not indicate the level or the orientation, and this should be indicated in panel b. The "holes" in panel c do not show crystalline cones, they show holes.

We clearly describe that this is a tangential section and from the other images it is evident what we see.

Yes, but in other regions these holes are filled, again this is evidence for the putative crystalline cones being morphologically independent units. This is a normal scientific deduction. Again, this panel is part of a number of related panels showing different perspectives. We do not demonstrate the existence of cones with the holes.

Panel d: this does not show cones but shows electron translucent areas that the authors assume were cones. But tetraconate cones reveal membrane borders demarcating their constituent cell borders. These are not shown here.

This criticism is not substantiated. This is a light microscopic image and the cones are not assumed by us, they are present as transparent round structures encircled by pigment cells as described in the captions.

Panel f: there are no obvious lenses in this figure, and the layer beneath the one indicating lenticular cuticle does not show cones but shows "holes" claimed as the absent cones: thus, an assumption, not a demonstration. Assuming an absence does not constitute evidence of past presence.

See above.

Panel g: assumptions are made that blurry profiles beneath a blurry surface must be cones. This is as unconvincing as the "artefactual cone layer" shown from *Fuxianhuia* in suppl. Fig. 1.

This is an original preparation of Lindström and it is a particular nice one (but not as thin as modern histological sections of fresh material, of course). The cones are clearly visible but open to interpretation as being cuticular or crystalline. We argue that the absence in some areas suggests the morphological independence. That is more than has been done in any other study on cones. We deleted the Fuxianhuia image, but we never claimed a cone layer being present in this figure – on the contrary.

Panel h: a section of the eye of the centipede *Scutigera purports* to show structures indicated in panel g. But the absence of any clear correspondence is compounded by the lack of a focused image in g.

See above.

Panel i does not provide any contribution to the cone question.

As mentioned in the text, this is a contribution to the cuticular surface.

Panel j: This image shows the eye surface (which is not indicated by the authors). Beneath the layer of lenses (here referred to as cuticle) arrows indicate absent cones. However, it is an act of faith that what are absent were once present.

As indicated in the captions it is a section through the eye. Concerning the holes, see above.

Panel k: this shows the assumed sloughing (exuvia) of the trilobite cuticle. However, the panel provides no supporting evidence for cones.

It is not an assumed exuvia, it is one. The internal layers of the cuticular cones are clearly visible and labelled.

Fig. 3. Panel a: The image has been taken at such low magnification that the transition of eye to head is not apparent. The asterisk may possibly indicate a basal membrane, but it could equally be a smooth fracture. If preservation was good enough to maintain features of photoreceptors (which are neurons) as claimed, then one would expect to see preserved retinula axons (these are neuronal prolongations of photoreceptors) extending further beneath the basement membrane. There is no evidence for this here.

Panel b inset: what is meant by a putative optical unit? The image suggests the inner faces of the cuticular lenses. It is unfortunate that the resolution is poor and the magnification low. Would it be sharper and at higher magnification, then if the "optical units" would reveal the inner face of the dioptric apparatus the observer might be able to see and count the tips of photoreceptors abutting the crystalline cone which is preserved should show its cruciform composition.

This criticism appears unjustified. We express every interpretation with great care. This should not be used to blame us for being sloppy. Furthermore, the reviewer asks for fine grained details and criteria that are used to doubt any of our interpretations, but it is a 400 million years old fossil and not a recent animal. What we do corresponds to the standard of other palaeontological studies. One should remember that the Schoenemann et al. 2017 and

the Strausfeld et al. 2016 studies (to use just two examples) have been carried out with a stereomicroscope. Panel b is meant to demonstrate that the superficially visible structures are repeated inside the rock. This scrutinizing of the argumentation should not be turned against us. Other studies just use an examination of the superficial aspects with a stereomicroscope and this seems to be accepted by the reviewer.

Panel c; Here the fractured eye shows longitudinal subdivisions, the periodicity of which approximately corresponds to surface structures interpreted as lenses. However, arbitrarily labeling a level beneath the surface as crystalline cones may, to the authors, be where they expect to see cones but for the reader there is nothing to suggest a differentiated structure at that level. The image of crystalline cones of the extant eumalacostracan krill *M. norvegica* in panel d is no help in that it amplifies the absence of a corresponding structure in panel c.

As in Fig. 2, the panels have to be seen in context. Hence panel c is meant to present an overview, which is treated in more detail in the other panels.

Panels e, f, and g: these are the images pivotal to the authors' claim of a trilobite apposition eye. However, using the scale bars to compare the putative cones in the exposed eye depth (panel e) with the surface rendering (panel f) one arrives at a putative cone depth (the green area painted on panel f) of about 50 microns in e whereas the structure indicated as a cone in panel f would be less than half that. The image of the rhabdomeres in panel g, from the extant *M. norvegica*, is somewhat misleading. These photoreceptors are far removed from the dioptric layer due to the intervening clear zone (filled with column like structures, which elicit some awkward but obvious questions) of an eye that is an optical superposition eye, not an apposition eye. If the authors are claiming that the trilobite eye was an apposition eye why haven't they used a crustacean proxy with an apposition eye? Never mind. What is a most reasonable interpretation of a rhabdomeric structure in panel e appears to correspond to the layered arrangement of rhabdomeres in *M. norvegica*, However, despite the colors ascribed to what the authors assume are elements of an apposition retina, the claim that the dioptric apparatus of *A. warsteinensis* comprises a cuticular lens succeeded by a crystalline cone is not defined well enough to be convincing. The authors may have convinced themselves, but this reviewer cannot see any indication of a separate tetraconate crystalline cone disposed beneath, and distinct from, a clearly resolve inner face of a cuticular lens; nor, proximally, a crystalline cone's delineation from the apices of underlying photoreceptors. Such features would be required for differentiating a crystalline cone from deeply penetrating cuticular lenses.

Again the reviewer is not convinced, but how could he/she? We (and apparently the two other reviewers) think that there is enough evidence for our claims on internal structures. Again the reviewer sets up criteria that are not met by any of the previous studies. Our measurements of the cones in panels e and f leads to similar depths of the cones. The different perspectives (frontal view versus view from above) have to be taken into account.

Lines 153-156. "In particular, the preservation of the internal eye anatomy of the Jurassic thylacocephalan crustacean *Dollocaris ingens* reveals many details including crystalline cones and a rhabdom structure that resemble the pattern found in the trilobites studied here²⁰." This comment is an extraordinary distortion: if anything, the converse applies. The exquisite

preservation of the dioptric apparatus elements and rhabdoms of *D. ingens* sets the benchmark for any comparison. In Vannier et al.'s paper each element is unambiguously identified and delineated. It does not resemble "the pattern in the trilobites studied here" because there is no definitive pattern and only one trilobite species reveals anything that might approach a rhabdomic structure comparable to those illustrated from *D. ingens*.

We agree that the degree of preservation in Dollocaris is excellent. This is a lucky case but one must say that this fossil is much younger than the ones we studied. But the whole paragraph is no argument against our results. It is just an expression of opinions of the reviewer.

In my review of the original manuscript I pointed out that spending a lot of type criticizing other studies to shed a kinder light on one's own work was a strategy that can easily backfire. Yet instead of taking this advice to heart, the authors have further embroidered and extended an already lengthy polemic levelled against earlier studies of Cambrian eye morphologies. While Scholtz et al. are entitled to critical opinions, if these are submitted for publication then they have to stand up to scrutiny: are they accurate; are they informative; do they strengthen their own observations? The following selections illustrate that none of these criteria pertain.

From lines 184-210, the Discussion reads as a sober assessment of trilobite eye studies concluding with the author assertion that their study supports the conclusion of that of Schoenemann et al, in PNS 2017. In short, the data of the present account, if correct, would confirm Schoenemann et al's claim that the trilobite eye was an apposition eye type and because Schoenemann's species derives from the lower Cambrian (even deeper than the Chengjiang biota) it is likely to represent the earliest apposition retina.

Again the repeatedly used argument of age. See our comments above. To the best of our knowledge we map the eye structures of trilobites on the admittedly problematic minimal consensus of trilobite phylogeny and come to the conclusion that the strange eye of Schmidtiellus might be derived in many respects. The question of whether this was an apposition eye is only a side aspect to us.

From line 211 things begin to get decidedly strange, beginning with an obscure argument that some minor oddities of Schoenemann's early Cambrian specimen, such as widely separated facets, suggest that its apposition characters are evolutionarily derived. But derived from what? The reader is left uninformed. And if they were derived, why couldn't the trilobite eyes claimed in this paper as apposition type eyes likewise be derived?

This comment is difficult to accept. Derived is just another expression for apomorphic and its commonly used. What we compare is the structure and not so much the question of existence of apposition optics or not. The reviewer consequently mixes these two aspects. We think they have to be treated largely independently. Apposition eyes can have different structures.

But then the authors make the serious mistake, when discussing work on Chengjiang fossil eyes, in questioning why one species of "Megacheira" possess *Limulus* like cuticular cones whereas another species possessed crystalline cones for apposition optics. I will return to this point presently. The authors go on to ratchet up their argument by challenging fossil

preservation, the quality of the specimens, and more besides. I have selected just a few of these comments, not because they have anything to do with the matter at hand – namely trilobite eyes – but as a demonstration of why it is wiser to think carefully before setting up a straw man scenario. It’s a strategy that can seriously backfire, as it has here.

Lines 255-259. Scholtz et al write: “First of all it is astonishing that extremely detailed structures

such as crystalline cones, which are made up of just a few cells, should be preserved despite the complex fossilization process of the Chengjiang Lagerstätte that involves flattening, microbial degradation, chemical alteration and replacement³⁸.” This rebuke ignores the established fact that the trilobite has a calcareous exoskeleton that resists flattening, is impermeable and is therefore notorious for precluding internal fossilization. That being established, isn’t it the more “astonishing” that cone cells and rhabdomeres protected by a calcareous exoskeleton would nevertheless be fossilized? Despite the authors’ astonishment, they know that there is nothing astonishing about preservational processes providing soft tissue preservation in soft cuticle Chengjiang species. They cite Parry et al. (BioEssays 40, 2018) as a source reference for exceptional preservation. They have likely read Ma et al., 2014 describing taphonomic events providing exceptional preservation of Fuxianhuia, for example. However, Scholtz et al. underpin their “astonishment” by citing a paper by Liu et al 2018 that is devoid of any insights into the process of preservation but which instead trivially conflates the identification of organs with biofilms, the asymmetries of which disqualify their interpretation of soft tissue.

We do not agree with the reviewer’s opinion, but we deleted the entire part about the preservation.

Lines 273-275. The authors ask, “Why, for instance, did they (the authors of reference 3) interpret the cone-like structures found in the eye of *Fortiforceps foliosa* as crystalline cones and

those in *Leancoilia illecebrosa* as cuticular cones³?” That the authors ask this question is perplexing as there are two clades comprising fossils traditionally accorded to “Megacheira.” “Megacheira,” sometimes referred to as great appendage (GA) arthropods, is not only exemplified by species such as *Leancoilia illecebrosa* and *Allalcomenaeus* where two pairs of

eyes are set flush with the cuticle: it is this 2+2 eye type that possess cuticular type cones. In contrast, *Fortiforceps*, also historically grouped in “Megacheira,” is a member of a clade multisegmented GA arthropods. This second clade comprises species that lack the discrete tagmosis of *Leancoiliids* and which, despite having stubby pincer-like deutocerebral appendages, have just one pair of eyes, each eye surmounting an eye stalk, as do other multisegmented “Megacheira” e.g. *Pseudoiulia cambriensis* and *Jianfengia multisegmentalis*. There are obvious morphological distinctions of eye types belonging to GA arthropods (with two

distinct morphologies of GAs) that differentiate two distinct clades of “Megacheir” and the morphology of their deutocerebral appendages further differentiate Megacheira: those with double eyes possess long, delicate caliber-like “great appendages” whereas multisegmented Megacheira, with single pairs of compound eyes, possess short pincer-like GAs, as mentioned above. It is important that criticisms of published works do not suffer this sort of confusion.

We know what the differences between the various megacheirans are. However, this does not necessarily mean that their eyes are different as well. Moreover, in some of the recent cladistics analyses they are monophyletic in others they are not. Sometimes they are interpreted as stem lineage euarthropods in others they are interpreted as stem-lineage chelicerates. The reviewer insists on his/her view of the situation putting any alternative view down.

Lines 278. "...the phylogenetic position of megacheirans and fuxianhuiids is still controversial." The authors omit any citation in support of this statement, when the most recent phylogenies, such as Legg, D. A. Nat. Comm. 4, 2485 (2013), are available.

We do not understand this comment at all, since Legg et al. is cited in this place.

281 -284. "For instance, if indeed the megacheiran *Leanchoilia illecebrosa* possesses a chelicerate eye-type and if it is a stem lineage euarthropod then this renders the structure of the ommatidia of the xiphosuran eye plesiomorphic for crown-group euarthropods." Yes, as does the eye of *Radiodonta*, and one has no problem with that. However, this has no bearing on their thesis that retinas of two trilobites, one Ordovician and another Silurian, are representatives of a plesiomorphic condition of Euarthropoda that presumably originated in the lower Cambrian or before. Even if convincing, which the manuscript paper is not, descriptions of eyes of late evolving trilobite species can't serve as proxies for an ancestral morph. The Schoenemann et al. account does, however, being based on a specimen from around 520mya.

There is a fundamental misunderstanding. We never claimed that the trilobite eyes represent the plesiomorphic condition of Euarthropoda. The opposite is true. Furthermore, in any case the eyes of Radiodonta are phylogenetically older than those of the oldest trilobite, even if this species stems from an older sediment. The latter fact, does not mean that the Schmidtiellus eye is ancestral for radiodontans as well.

lines 285-287 ".....there is no indication that the eyes of *Limulus* underwent loss of a crystalline cone, since there are no cells corresponding to cone cells...." Here the authors are referring to a paper by Fahrenbach (reference 7) but not cited at this juncture. Reference 7 discusses Semper cell-like cone cells. In my first review, I drew attention to the second Fahrenbach paper, which is here totally ignored, but which describes *Limulus* cone cells. Let me cite the 1969 paper again: "Fahrenbach, W. H. 1969. The morphology of the eyes of *Limulus* II. Ommatidia of the compound eye. *Z. Zellforsch* 93. 451-483). That cones cells are present in *Limulus* negates Scholtz et al.'s contention that the *Limulus* type eye (or the eyes of *Alalcomenaeus* or *Leanchoilia*) cannot have derived from a plesiomorphic apposition ground pattern established in deep time.

We replied to this argument about the Fahrenbach articles above.

It is unfortunate that the supplemental figure 1 has been used to argue that inaccurate interpretations have, in the past, been published relating to the eye morphology in the species *Fuxianhuia*. The specimen in suppl. Fig. 1, also *Fuxianhuia*, is illuminated very obliquely with much back-scatter allowing great latitude of interpretations. It appears that the specimen is from a private collection, lacking an accession number which would need to be obtained if the intention is to publish it in a journal.

A direct comparison of the Suppl. Fig. 1 and those published before reveal striking similarities. Likewise, the shadows of the published figures show that the specimen was also illuminated obliquely. This is justified because otherwise one would not see anything. An accession number was mentioned in the Material and Method section. Nevertheless, we removed this figure.

Reviewers' Comments:

Reviewer #1:

Remarks to the Author:

The authors have dealt with my suggestions from previous rounds of review. They have done a solid job documenting their material and interpreting it in terms of the standard picture of compound eye evolution in extant arthropods, and adequately related their work to existing data in other fossils. This version has benefited from the review process, e.g., explaining how the different preservational modes are consistent with the lens and cone being independent structures. In my opinion, the questions posed here and the novelty of the findings about the fine structure of trilobite eyes were always of sufficiently importance for the journal.

It would be good to get the Archegonus material registered in the MfN collection (with a cited registration number in the paper) rather than just "will be deposited".

REVIEWERS' COMMENTS:

Reviewer #1 (Remarks to the Author):

The authors have dealt with my suggestions from previous rounds of review. They have done a solid job documenting their material and interpreting it in terms of the standard picture of compound eye evolution in extant arthropods, and adequately related their work to existing data in other fossils. This version has benefited from the review process, e.g., explaining how the different preservational modes are consistent with the lens and cone being independent structures. In my opinion, the questions posed here and the novelty of the findings about the fine structure of trilobite eyes were always of sufficiently importance for the journal.

It would be good to get the *Archegonus* material registered in the MfN collection (with a cited registration number in the paper) rather than just “will be deposited”.

Done